# POMRL: No-Regret Learning-to-Plan with Increasing Horizons

**Khimya Khetarpal**$^{*\dagger}$                                    *khimya.khetarpal@mail.mcgill.ca*
*Department of Computer Science*
*McGill University*

**Claire Vernade**$^{*\ddagger}$                                    *claire.vernade@gmail.com*
*University of Tuebingen*

**Brendan O'Donoghue**                                    *bodonoghue@google.com*
*Google Deepmind*

**Satinder Singh Baveja**                                    *baveja@google.com*
*Google Deepmind*

**Tom Zahavy**                                    *tomzahavy@google.com*
*Google Deepmind*

**Reviewed on OpenReview:** *https://openreview.net/forum?id=brGgOAXYtr*

## Abstract

We study the problem of planning under model uncertainty in an online meta-reinforcement learning (RL) setting where an agent is presented with a sequence of related tasks with limited interactions per task. The agent can use its experience in each task *and* across tasks to estimate both the transition model and the distribution over tasks. We propose an algorithm to meta-learn the underlying relatedness across tasks, utilize it to plan in each task, and upper-bound the regret of the planning loss. Our bound suggests that the average regret over tasks decreases as the number of tasks increases and as the tasks are more similar. In the classical single-task setting, it is known that the planning horizon should depend on the estimated model's accuracy, that is, on the number of samples within task. We generalize this finding to meta-RL and study this dependence of planning horizons on the number of tasks. Based on our theoretical findings, we derive heuristics for selecting slowly increasing discount factors, and we validate its significance empirically.

## 1 Introduction

*Meta-learning* (Caruana, 1997; Baxter, 2000; Thrun & Pratt, 1998; Finn et al., 2017; Denevi et al., 2018) offers a powerful paradigm to leverage past experience to reduce the sample complexity of learning future related tasks. *Online meta-learning* considers a sequential setting, where the agent progressively accumulates knowledge and uses past experience to learn good priors and to quickly adapt within each task Finn et al. (2019); Denevi et al. (2019). Robots acting in real world for instance need to be responsive to and robust against perturbation inherent in the environment dynamics and their decision making. When the tasks share a structure i.e. have similar transition dynamics and are related, such approaches enable progressively faster convergence, or equivalently better model accuracy with better sample complexity (Schmidhuber & Huber, 1991; Thrun & Pratt, 1998; Baxter, 2000; Finn et al., 2017; Balcan et al., 2019).

---

$^{*}$Equal Contribution
$^{\dagger}$Work partially done during an internship at DeepMind. Now at Deepmind.
$^{\ddagger}$Work partially done at DeepMind.

In model-based reinforcement learning (RL), the agent uses an estimated model of the environment to plan actions ahead towards the goal of maximizing rewards. A key component in the agent's decision making is the horizon used during planning. In general, an *evaluation horizon* is imposed by the task itself, but the learner may want to use a different and potentially shorter *guidance horizon*. In the discounted setting, the size of the evaluation horizon is of order $(1 - \gamma_{\texttt{eval}})^{-1}$, for some discount factor $\gamma_{\texttt{eval}} \in (0, 1)$, and the agent may use $\gamma \neq \gamma_{\texttt{eval}}$ for planning. For instance, a classic result known as Blackwell Optimality (Blackwell, 1962) states there exists a discount factor $\gamma^\star$ and a corresponding optimal policy such that the policy is also optimal for any greater discount factor $\gamma \geq \gamma^\star$. Thus, an agent that plans with $\gamma = \gamma^\star$ will be optimal for any $\gamma_{\texttt{eval}} > \gamma^\star$. In the Arcade Learning Environment (Bellemare et al., 2013) a discount factor of $\gamma_{\texttt{eval}} = 1$ is used for evaluation, but typically a smaller $\gamma$ is used for training (Mnih et al., 2015). Using a smaller discount factor acts as a regularizer (Amit et al., 2020; Petrik & Scherrer, 2008; Van Seijen et al., 2009; François-Lavet et al., 2019; Arumugam et al., 2018) and reduces planner over-fitting in random MDPs (Arumugam et al., 2018). Indeed, the choice of planning horizon plays a significant role in computation (Kearns et al., 2002), optimality (Kocsis & Szepesvári, 2006), and on the complexity of the policy class (Jiang et al., 2015). In addition, meta-learning discount factors has led to significant improvements in performance (Xu et al., 2018; Zahavy et al., 2020; Flennerhag et al., 2021; 2022; Luketina et al., 2022).

When doing model-based RL with a learned model, the optimal guidance planning horizon, called *effective horizon* by Jiang et al. (2015), depends on the accuracy of the model, and so on the amount of data used to estimate it. Jiang et al. (2015) show that when data is scarce, a guidance discount factor $\gamma < \gamma_{\texttt{eval}}$ should be preferred for planning. The reason for this is straightforward; if the model used for planning is inaccurate, then errors will tend to accumulate along the planned trajectory. A shorter effective planning horizon will accumulate less error and may lead to better performance, even when judged using the true $\gamma_{\texttt{eval}}$. While that work treated only the batch, single-task setting, the question of effective planning horizon remains open in the online meta-learning setting where the agent accumulates knowledge from many tasks, with limited interactions within each task.

In this work, we consider a *meta-reinforcement-learning* problem made of a sequence of **related tasks**. We leverage this structural task similarity to obtain model estimators with faster convergence as more tasks are seen. The central question of our work is:

> *Can we meta-learn the model across tasks and adapt the effective planning horizon accordingly?*

We take inspiration from the *Average Regret-Upper-Bound Analysis* [ARUBA] framework (Khodak et al., 2019) to generalize planning loss bounds to the meta-RL setting. A high-level, intuitive outline of our approach is presented in Fig. 1. **Our main contributions** are as follows:

- We formalize planning in a model-based meta-RL setting as an *average planning loss* minimization problem, and we propose an algorithm to solve it.
- Under a structural *task-similarity* assumption, we prove a novel high-probability task-averaged regret upper-bound on the planning loss of our algorithm, inspired by ARUBA. We also demonstrate a way to learn the task-similarity parameter $\sigma$ on-the-fly. To the best of our knowledge, this is a first formal (ARUBA-style) analysis to show that meta-RL can be more efficient than RL.
- Our theoretical result highlights a new dependence of the planning horizon on the size of the within-task data $m$ *and* on the number of tasks $T$. This observation allows us to propose two heuristics to adapt the planning horizon given the overall sample-size.

## 2 Preliminaries

**Reinforcement Learning.** We consider tabular Markov Decision Processes (MDPs) $\mathcal{M} = \langle \mathcal{S}, \mathcal{A}, R, P, \gamma_{\texttt{eval}} \rangle$, where $\mathcal{S}$ is a finite set of states, $\mathcal{A}$ is a finite set of actions and we denote the set cardinalities as $S = |\mathcal{S}|$ and $A = |\mathcal{A}|$. For each state $s \in \mathcal{S}$, and for each available action $a \in \mathcal{A}$, the probability vector $P(\cdot \mid s, a)$ defines a transition model over the state space and is a probability distribution in a set of feasible models $\mathcal{D}_P \subset \Delta_S$, where $\Delta_S$ the probability simplex of dimension $S - 1$. We denote $\Sigma \leq 1$ the diameter of $\mathcal{D}_P$. A policy is a function $\pi : \mathcal{S} \to \mathcal{A}$ and it characterizes the agent's behavior.

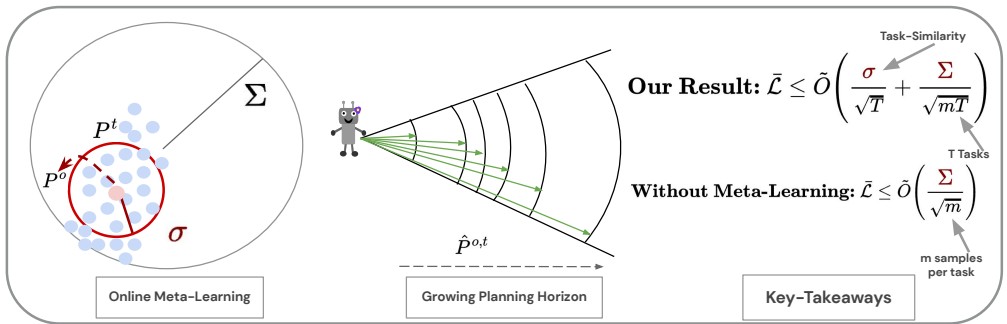

Figure 1: **Effective Planning Horizons in Meta-Reinforcement Learning.** The agent faces a sequence of tasks with transition vector $(P^t)_{t \in [T]}$ (probability vectors represented by blue dots) all close to each other ($\sigma < \Sigma = 1$). The agent builds a transition model for each task and plans with these inaccurate models. By using data from previous tasks, the agent meta-learns an initialization of the model ($\hat{P}^{o,t}$), which leads to better planning in new related but unseen tasks. We show an improved average regret upper bound that scales with task-similarity parameter $\sigma$ and inversely with the number of tasks $T$: as knowledge accumulates, uncertainty diminishes, and the agent can plan with longer horizons. All tasks $P^t \sim \mathcal{P}$ are centered at some fixed but unknown $P^o$, depicted here by the shaded red dot and pointed by the arrow.

We consider the bounded reward setting, *i.e.*, $R \in [0, R_{\max}]$ and without loss of generality we set $R_{\max} = 1$ (unless stated otherwise). Given an MDP, or task, $M$, for any policy $\pi$, let $V_{M,\gamma}^{\pi} \in \mathbb{R}^S$ be the value function when evaluated in MDP $M$ with discount factor $\gamma \in (0,1)$ (potentially different from $\gamma_{\texttt{eval}}$); defined as $V_{M,\gamma}^{\pi}(s) = \mathbb{E} \sum_{t=0}^{\infty} (\gamma^t R_{s_t} \mid s_0 = s)$. The goal of the agent is to find an optimal policy, $\pi_{M,\gamma}^{\star} = \arg\max_{\pi} \mathbf{E}_{s \sim \rho} V_{M,\gamma}^{\pi}(s)$ where $\rho > 0$ is any positive measure, denoted $\pi^{\star}$ when there is no ambiguity. For given state and action spaces and reward function $(\mathcal{S}, \mathcal{A}, R)$, we denote $\Pi_{\gamma}$ the set of *potentially* optimal policies for discount factor $\gamma$: $\Pi_{\gamma} = \{\pi \mid \exists P \text{ s.t. } \pi = \pi_{M,\gamma}^{\star} \text{ where } M = \langle \mathcal{S}, \mathcal{A}, R, P, \gamma \rangle \}$. We use Big-O notation, $O(\cdot)$ and $\tilde{O}(\cdot)$, to hide respectively universal constants and poly-logarithmic terms in $T, S, A$ and $\delta > 0$ (the confidence level).

**Model-based Reinforcement Learning.** In practice, the true model of the world is unknown and must be estimated from data. One approach to approximately solve the optimization problem above is to construct a model, $\langle \hat{R}, \hat{P} \rangle$ from data, then find $\pi_{\hat{M},\gamma}^{\star}$ for the corresponding MDP $\hat{M} = \langle \mathcal{S}, \mathcal{A}, \hat{R}, \hat{P}, \gamma \rangle$. This approach is called *model-based RL* or *certainty-equivalence (CE) control*.

**Planning with inaccurate models.** In this setting, Jiang et al. (2015) define the planning loss as the gap in expected return in MDP $M$ when using $\gamma \leq \gamma_{\texttt{eval}}$ and the optimal policy for an approximate model $\hat{M}$:

$$\mathcal{L}(\hat{M}, \gamma \mid M, \gamma_{\texttt{eval}}) = \|V_{M,\gamma_{\texttt{eval}}}^{\pi_{M,\gamma_{\texttt{eval}}}^{\star}} - V_{M,\gamma_{\texttt{eval}}}^{\pi_{\hat{M},\gamma}^{\star}}\|_{\infty}.$$

Thus, the **optimal effective planning horizon** $(1-\gamma^{\star})^{-1}$ is defined using the discount factor that minimizes the planning loss, *i.e.*, $\gamma^{\star} := \min_{0 \leq \gamma \leq \gamma_{\texttt{eval}}} \mathcal{L}(\hat{M}, \gamma \mid M, \gamma_{\texttt{eval}})$.

**Theorem 1.** *(Jiang et al. (2015)) Let $M$ be an MDP with non-negative bounded rewards and evaluation discount factor $\gamma_{\texttt{eval}}$. Let $\hat{M}$ be the approximate MDP comprising the true reward function of $M$ and the approximate transition model $\hat{P}$, estimated from $m > 0$ samples for each state-action pair. Then, with probability at least $1 - \delta$,*

$$\left\|V_{M,\gamma_{eval}}^{\pi_{M,\gamma_{eval}}^{\star}} - V_{M,\gamma_{eval}}^{\pi_{\hat{M},\gamma}^{\star}}\right\|_{\infty} \leq \frac{\gamma_{eval} - \gamma}{(1 - \gamma_{eval})(1 - \gamma)} + \frac{2\gamma R_{\max}}{(1 - \gamma)^2} \left( \sqrt{\frac{\Sigma}{2m} \log \frac{2SA|\Pi_{\gamma}|}{\delta}} \right) \tag{1}$$

*where $\Sigma$ is upper-bounded by 1 as $P, \hat{P} \in \Delta_S$.*

This result holds for a count-based model estimator (i.e, empirical average of observed transitions) given by a generator model for each pair $(s, a)$. It gives an upper-bound on the planning loss as a function of the

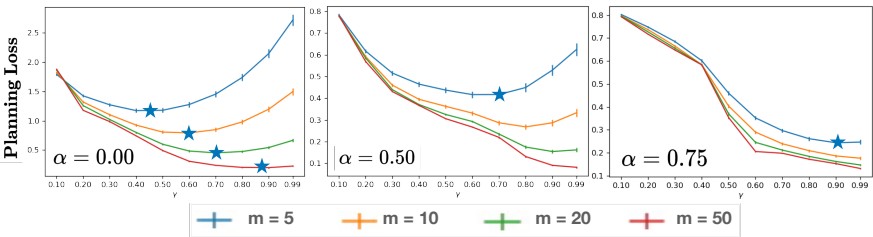

Figure 2: **On the role of incorporating a ground truth prior of transition model on planning horizon.** The planning loss is a function of the discount factor $\gamma$ and is impacted by incorporating prior knowledge. The learner has $m = 5, 10, 20, 50$ samples per task to estimate the model, corresponding to the curves in each sub figure. Inspecting any of the sub figures, we observe that larger values of $m$ lead to lower planning loss and a larger effective discount factor. Besides, inspecting one value of $m$ across tasks (e.g., $m = 5$ ), we see that the same effect (lower planning loss and larger effective discount) occurs when the learner puts more weight on the ground truth prior through $\alpha$.

guidance discount factor $\gamma < 1$. The result decomposes the loss into two terms: the constant bias which decreases as $\gamma$ tends to $\gamma_{\texttt{eval}}$, and the variance (or uncertainty) term which increases with $\gamma$ but decreases as $1/\sqrt{m}$. As $m \to \infty$ that second factor vanishes, but in the low-sample regime the optimal effective planning horizon should trade-off both terms.

**Illustration.** These effects are illustrated in Fig. 2 on a simple 10-state, 2-action random MDP. The leftmost plot uses the simple count-based model estimator and reproduces the results from Jiang et al. (2015). We then incorporate the true prior (mean model $P^o$ as in Fig 1 and defined above Eq. 3 in Assumption 1) in the estimator with a growing mixing factor $\alpha \in (0,1)$: $\hat{P}(m) = \alpha P^o + (1 - \alpha)\frac{\sum^i X^i}{m}$. We observe that increasing the weight $\alpha \in (0,1)$ on good prior knowledge enables longer planning horizons and lower planning loss.

**Online Meta-Learning and Regret.** We consider an online meta-RL problem where an agent is presented with a sequence of tasks $M_1, M_2, ..., M_T$, where for each $t \in [T]$, $M_t = \langle \mathcal{S}, \mathcal{A}, P^t, R, \gamma_{\texttt{eval}}\rangle$, that is, the MDPs only differ from each other by the transition matrix (dynamics model) $P^t$. The learner must sequentially estimate the model $\hat{P}^t$ for each task $t$ from a batch of $m$ transitions simulated for each state-action pair[1].

Its goal is to minimize the average planning loss also expressed in the form of task averaged regret suffered in planning and defined as

$$\bar{\mathcal{L}}(\hat{M}_{1:T}, \gamma | M_{1:T}, \gamma_{\texttt{eval}}) = \frac{1}{T}\sum_{t=1}^{T} \mathcal{L}(\hat{M}_t, \gamma | M_t, \gamma_{\texttt{eval}}) = \frac{1}{T}\sum_{t=1}^{T} \|V_{M_t, \gamma_{\texttt{eval}}}^{\pi^{\star}_{\hat{M}_t, \gamma_{\texttt{eval}}}} - V_{M_t, \gamma_{\texttt{eval}}}^{\pi^{\star}_{\hat{M}_t, \gamma}}\|_\infty \tag{2}$$

Note that the reference MDP for each term is the true $M_t$, and the discount factor $\gamma$ is the same in all tasks. One can see this objective as a stochastic dynamic regret: at each task $t \in [T]$, the learner competes against the optimal policy for the *current* true model, as opposed to competing against the best fixed policy in hindsight used in classical definitions of regret.

Note that **our dynamic regret is different from the one considered in ARUBA** (Khodak et al., 2019). They consider the fully online setting where the data is observed as an arbitrary stream within each task, and each comparator is simply the minimum of the within-task loss in hindsight. In our model, however, given access to a simulator (See Sec. 2) allows us to get i.i.d transition samples as a batch at the beginning of each task, and consequently we define our regret with respect to the true generating parameter. One key consequence of this difference is that their regret bounds cannot be directly applied to our setting, and we prove new results further below.

---

[1]So a total of $mSA$ samples.

# 3 Planning with Online Meta-Reinforcement Learning

We here formalize planning in a model-based meta-RL setting. We start by specifying all our assumptions in Sec 3.1 including our main assumption about task relatedness in Sec. 1, present our approach and explain the proposed algorithms POMRL and ada-POMRL in Sec. 3.2. Our main result is a high-probability upper bound on the average planning loss under the assumed task relatedness, presented as Theorem 2.

## 3.1 Assumptions

In many real world scenarios such as robotics, it is required to be responsive to changes in the environment and, at the same time, to be robust against perturbation inherent in the environment and their decision making. In such practical scenarios, the key reason to employ meta-learning is for the learner to leverage **task-similarity** (or task variance) across tasks. Bounded task similarity is becoming a core assumption in the analysis of recent meta learning (Khodak et al., 2019) and multi-task (Cesa-Bianchi et al., 2021) online learning algorithms.

**Assumption 1** (Structural Assumption Across Tasks: Task Relatedness)**.** *In this work, we exploit the structural assumption that for all $t \in [T]$, $P^t \sim \mathcal{P}$ centered at some fixed but unknown $P^o \in \Delta_S^{S \times A}$ and such that for any $(s,a)$,*

$$\|P_{s,a}^t - P_{s,a}^o\|_\infty \le \sigma = \max_{(s,a)} \sigma(s,a) \quad a.s. \tag{3}$$

This also implies that $\max_{t,t'} \|P_{s,a}^t - P_{s,a}^{t'}\|_\infty \le 2\sigma$, and that the meta-distribution $\mathcal{P}$ is bounded within a small subset of the simplex. It is immediate to extend our results under a high-probability assumption instead of the almost sure statement above. In our experiments, we will use Gaussian or Dirichlet priors over the simplex, whose moments are bounded with high-probability, not almost surely. Importantly, we will say that a multi-task environment is *strongly structured* when $\sigma < \Sigma$, *i.e.* when the effective diameter of the models is smaller than that of the entire feasible space.

**Assumption 2** (Access to a Simulator)**.** *We assume that for each task $t \in [T]$ we have access to a simulator of transitions (Kearns et al., 2002) providing $m$ i.i.d. samples $(X_{s,a}^{t,i})_{i=1..m} \in \mathcal{S}^m \sim P^t(\cdot|s,a)$ (categorical distribution).*

Next, for simplicity we assume throughout that the rewards are known and focus on learning and planning with an approximate dynamics model. Additionally estimating the reward is a straightforward extension of our analysis and would not change the implications of our main result.

**Assumption 3** (Known Rewards)**.** *Given a distribution of tasks, we assume that the rewards are known.*

## 3.2 Our Approach

With access to a simulator (Assumption 2); for each $(s,a)$, we can compute an empirical estimator for each $s' \in [S]$: $\bar{P}_{s,a}^t(s') = \sum_{i=1}^m \mathbb{1}\{X_{s,a}^{t,i} = s'\}/m$, with naturally $\sum_{s'} \bar{P}_{s,a}^t(s') = 1$. We perform meta-RL via alternating minimizing a batch *within-task* regularized least-squares loss, and an outer-loop step where we optimize the regularization to optimally balance bias and variance of the next estimator.

**Estimating dynamics model via regularized least squares.** We adapt the standard technique of meta-learned regularizer (see e.g. Baxter (2000); Cella et al. (2020) for supervised learning and bandit respectively) to this model estimation problem. At each round $t$, the **current model** $\hat{P}_{(s,a)}^t$ is estimated by minimizing **a regularized least square loss**: for a given **regularizer** $h_t$ (to be specified below)[2] and parameter $\lambda_t > 0$ for each $(s,a) \in \mathcal{S} \times \mathcal{A}$ we solve

$$\hat{P}_{(s,a)}^t = \underset{P_{(s,a)} \in \Delta_S}{\arg\min} \left\| \underbrace{\frac{1}{m}\sum_{i=1}^m \mathbb{1}\{X_{s,a}^{t,i}\}}_{\text{empirical transition prob.}} - P_{(s,a)} \right\|_2^2 + \lambda_t \|P_{(s,a)} - h_t\|_2^2, \tag{4}$$

---

[2]In principle, this loss is well defined for any regularizer $h_t$ but we specify a meta-learned one and prove that it induces good performance.

where we use $\mathbb{1}\{X^{t,i}_{s,a}\}$ to denote the one-hot encoding of the state into a vector in $\mathbb{R}^S$. Importantly, $h_t$ and $\lambda_t$ are meta-learned in the outer-loop (see below) and affect the bias and variance of the resulting estimator. The solution of equation 4 can be computed in closed form as a convex combination of the empirical average (count-based) and the prior: $\hat{P}^t = \alpha_t h_t + (1 - \alpha_t)\bar{P}^t$ where $\alpha_t = \frac{\lambda_t}{1+\lambda_t}$ is the current mixing parameter.

**Outer-loop: Meta-learning the regularization.** At the beginning of task $1 < t \le T$, the learner has already observed $t - 1$ *related but different* tasks. We define $h_t$ as an **average of Means (AoM)**:

$$h^t_{(s,a)} \leftarrow \hat{P}^{o,t}_{(s,a)} = \frac{1}{t-1} \sum_{j=1}^{t-1} \frac{\sum_{i=1}^m \mathbb{1}\{X^{j,i}_{(s,a)}\}}{m} := \frac{1}{t-1} \sum_{j=1}^{t-1} \bar{P}^j_{(s,a)}. \tag{5}$$

**Deriving the mixing rate.** To set $\alpha_t$, we compute the Mean Squared Error (MSE) of $\hat{P}^t_{(s,a)}$, and minimize an upper bound (see details in Appendix B): $\text{MSE}(\hat{P}^t_{(s,a)}) \le \alpha_t^2 \sigma^2 (1 + \frac{1}{t}) + (1 - \alpha_t)^2 \frac{1}{m}$, which leads to $\alpha_t = \frac{1}{\sigma^2(1+1/t)m+1}$.

**Algorithm** 1 depicts the complete pseudo code. We note here that POMRL $(\sigma)$ assumes, for now, that the underlying task-similarity parameter $\sigma$ is known, and we discuss a fully empirical extension further below (See Sec. 4). The learner does not know the number of tasks a priori and tasks are faced sequentially online. The learner performs meta-RL alternating between within-task estimation of the dynamics model $\hat{P}^t$ via a batch of $m$ samples for that task, and an outer loop step to meta-update the regularizer $\hat{P}^{o,t+1}$ alongside the mixing rate $\alpha_{t+1}$. For each task, we use a $\gamma$-Selection-Procedure to choose planning horizon $\gamma^* \le \gamma_{\texttt{eval}}$. We defer the details of this step to Sec. 6 as it is non-trivial and only a partial consequence of our theoretical analysis. Next, the learner performs planning with an imperfect model $\hat{P}^t$. For planning, we use dynamic programming, in particular policy iteration (a combination of policy evaluation, and improvement), and value iteration to obtain the optimal policy $\pi^\star_{\hat{P}^t,\gamma^*}$ for the corresponding MDP $\hat{M}_t$.

---

**Algorithm 1:** POMRL $(\sigma)$ – Planning with Online Meta-Reinforcement Learning

**Input:** Given task-similarity $(\sigma(s,a))$ a matrix of size $S \times A$. Initialize $\hat{P}^{o,1}$ to uniform, $\alpha_1 = 0$.
**for** *task* $t \in [T]$ **do**
    **for** $t^{th}$ *batch of m samples* **do**
        $\hat{P}^t(m) = (1 - \alpha_t)\frac{1}{m}\sum_{i=1}^m X_i + \alpha_t \hat{P}^{o,t}$    // regularized least squares minimizer.
        $\gamma^\star \leftarrow \gamma\text{-Selection-Procedure}(m, \alpha_t, \sigma, T, S, A)$
        $\pi^\star_{\hat{P}^t,\gamma^*} \leftarrow \texttt{Planning}(\hat{P}^t(m))$   //
        **Output:** $\pi^\star_{\hat{P}^t,\gamma^*}$
    Update $\hat{P}^{o,t+1}, \alpha_{t+1} = \frac{1}{\sigma^2(1+1/t)m+1}$   // meta-update AoM (Eq. 5) and mixing rate

---

### 3.3 Average Regret Bound for Planning with Online-meta-learning

Our main theoretical result below controls the average regret of POMRL $(\sigma)$, a version of Alg. 1 with additional knowledge of the underlying task relatedness, *i.e.*, the true $\sigma > 0$.

**Theorem 2.** *Using the notation of Theorem 1, we bound the average planning loss equation 2 for* POMRL $(\sigma)$*:*

$$\bar{\mathcal{L}} \le \frac{\gamma_{eval} - \gamma}{(1 - \gamma_{eval})(1 - \gamma)} + \frac{2\gamma S}{(1 - \gamma)^2}\tilde{O}\left(\frac{\sigma + \sqrt{\frac{1}{T}\left(\sigma + \sqrt{\sigma^2 + \frac{\Sigma}{m}}\right)}}{\sigma^2 m + 1} + \frac{\sigma^2 m\sqrt{\frac{\Sigma}{m}}}{\sigma^2 m + 1}\right) \tag{6}$$

*with probability at least $1 - \delta$, where $\sigma^2 < 1$ is the measure of the task-similarity and $\sigma = \max_{(s,a)} \sigma(s,a)$.*

The proof of this result is provided in Appendix D and relies on a new concentration bound for the meta-learned model estimator. The last term on the r.h.s. corresponds to the uncertainty on the dynamics.

First we verify that if $T = 1$ and $m$ grows large, the second term dominates and is equivalent to $\tilde{O}(\sqrt{\frac{\Sigma}{m}})$ (as $\sigma^2/(\sigma^2 m + 1) \to 0$), which is similar to that of Jiang et al. (2015) as there is no meta-learning, with an additional $O(\frac{1}{m})$ but second order term due to the introduced bias. Then, if $m$ is fixed and small, for small enough values of $\sigma^2$ (typically $\sigma < 1/\sqrt{m}$), the first term dominates and the r.h.s. boils down to $\tilde{O}\left((\sigma + \frac{1}{\sqrt{m}})/\sqrt{T}\right)$. This highlights the interplay of our structural assumption parameter $\sigma$ and the amount of data $m$ available at each round. The regimes of the bound for various similarity levels are explored empirically in Sec. 5 (Q3). We also show the dependence of the regret upper bound on $m$ and $T$ for a fixed $\sigma$, in Appendix Fig. F3.

**Implications for degree of task-similarity *i.e.*, $\sigma$ values.** Our bound suggests that the degree of improvement you can get from meta learning scales with the task similarity $\sigma$ instead of the set size $\Sigma$. Thus, for $\sigma \leq \Sigma$, performing meta learning with Algorithm 1 guarantees better learning measured via our improved regret bound when there is underlying structure in the problem space which we formalize through Eq. 3. Should $\sigma$ be large, the techniques will still hold and our bounds will simply scale accordingly.

**When $\sigma = 0$, all tasks are exactly the same.** Indeed, the mixing rate $\alpha_t \approx 1$ for all $t$, so our algorithm boils down to returning the average of means $\hat{P}^{o,t}$ for each task, which simply corresponds to solving the tasks as a continuous, uninterrupted stream of batches from the nearly same model that $\hat{P}^{o,t}$ aggregates. Unsurprisingly, our bound recovers that of (Jiang et al., 2015, Theorem 1): the bound below reflects that we have to estimate only one model in a space of "size" $\Sigma$ with $mT$ samples.

$$\bar{\mathcal{L}} \leq \frac{\gamma_{\texttt{eval}} - \gamma}{(1 - \gamma_{\texttt{eval}})(1 - \gamma)} + \frac{2\gamma S}{(1-\gamma)^2}\tilde{O}\left(\sqrt{\frac{\Sigma}{mT}}\right) \tag{7}$$

**When $\sigma = 1$, then $\sigma = \Sigma = 1$, then the meta-learning assumption is not relevant but our bound remains valid and gracefully degrades to reflect it.** We need to estimate $T$ models each with $m$ samples. Then the second term $\frac{1}{\sqrt{m}}$ reflects the usual estimation error for each task while the first term is an added bias (second order in $\frac{1}{m}$) due to our regularization to our mean prior $P^o$ that is not relevant here.

$$\bar{\mathcal{L}} \leq \frac{\gamma_{\texttt{eval}} - \gamma}{(1 - \gamma_{\texttt{eval}})(1 - \gamma)} + \frac{2\gamma S}{(1-\gamma)^2}\tilde{O}\left(\frac{1}{m}\left(1 + \frac{1}{\sqrt{T}}(1 + \sqrt{1 + \frac{1}{m}})\right) + \frac{1}{\sqrt{m}}\right) \tag{8}$$

**Connections to ARUBA.** As explained earlier, our metric is not directly comparable to that of ARUBA (Khodak et al., 2019) but it is interesting to make a parallel with the high-probability average regret bounds proved in their Theorem 5.1. They also obtain an upper bound in $\tilde{O}(1/\sqrt{m} + 1/\sqrt{mT})$ if one upper bounds their average within-task regret $\bar{U} \leq B\sqrt{m}$.

**Remark 1** (Role of the task similarity $\sigma$ in Eq. 2). *When $\sigma > 0$, POMRL naturally integrates each new data batch into the model estimation. The knowledge of $\sigma$ is necessary to obtain this exact and intuitive update rule, and our theory only covers POMRL equipped with this prior knowledge, but we discuss how to learn and plug-in $\hat{\sigma}_t$ in practice. Note that it would be possible to extend our result to allow for using the empirical variance estimator with tools like the Bernstein inequality, but we believe this it out of the scope of this work as it would essentially give a similar bound as obtained in Theorem 2 with an additional lower order term in $O(1/T)$, and it would not provide much further intuition on the meta-planning problem we study.*

## 4 Practical Considerations: Adaption On-The-Fly

In this section we propose a variant of POMRL that meta learns the task similarity parameter, which we call ada-POMRL . We compare the two algorithms empirically in a 10 state, 2 action MDP with closely related tasks with a total of $T = 15$ tasks (details of the experiment setup are deferred to Sec. 5).

**Performance of POMRL .** Recall that POMRL is primarily learning the regularizer and assumes the knowledge of the underlying task similarity (i.e. $\sigma$). We observe in Fig. 3 that with each round $t \in T$ POMRL is able to plan better as it learns and adapts the regularizer to the incoming tasks. The convergence rate and final performance corroborates with our theory.

**Can we also meta-learn the task-similarity parameter?** In practice, the parameter $\sigma$ may not be known and must be estimated online and plugged in (see Appendix C for details).

Alg. 2 `ada-POMRL` uses Welford's algorithm to compute an online estimate of the variance after every task using the model estimators, and simply plugs-in this estimate wherever `POMRL` was using the true value. From the perspective of `ada-POMRL` , `POMRL` is an "oracle", i.e. the underlying task-similarity is known. However, in most practical scenarios, the learner does not have this information a priori.

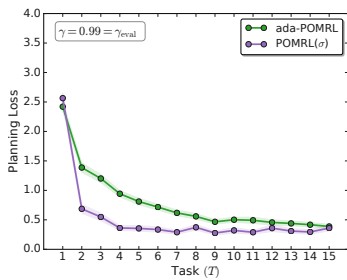

We compare empirically `POMRL` and `ada-POMRL` on a strongly structured problem ($\sigma \approx 0.01$) in Fig. 3 and observe that meta-learning the underlying task relatedness allows `ada-POMRL` to adapt to the incoming tasks accordingly. Adaptation on-the-fly with `ada-POMRL` comes at a cost *i.e.*, the performance gap in comparison to `POMRL` but eventually converges albeit with a slower rate. This is intuitive and a similar theoretical guarantee applies (See Remark 1).

Figure 3: `ada-POMRL` enables meta-learning the task-similarity on-the-fly with a performance gap for the initial set of tasks as compared to the oracle `POMRL` , but improves with more tasks

This online estimation of $\sigma$ means that `ada-POMRL` now requires an initial value for $\hat{\sigma}_1$, which is a choice left to the practitioner, but will only affect the results of a finite number of tasks at the beginning. Using $\hat{\sigma}_1$ too small will give a slightly increased weight to the prior in initial tasks, which is not desirable as the latter is not yet learned and will result in an increased bias. On the other hand, setting $\hat{\sigma}_1$ too large (i.e close to $1/2$) will decrease the weight of the prior and increase the variance of the returned solution; in particular, in cases where the true $\sigma$ is small, a large initialization will slow down convergence and we observe empirical larger gaps between `POMRL` and `ada-POMRL` . In the extreme case where $\sigma \approx 0$, a large initialization will drastically slow down `ada-POMRL` as it will take many tasks before it *discovers* that the optimal behavior is essentially to aggregate the batches.

---

**Algorithm 2:** `ada-POMRL` – Planning with Online Meta-Reinforcement Learning

---

**Input:** Initialize $\hat{P}^{o,1}$ to uniform, $(\hat{\sigma})_1$ as a matrix of size $S \times A$,$\alpha_1 = 0$.
**for** *task* $t \in [T]$ **do**

    **for** $t^{th}$ *batch of m samples* **do**

        $\hat{P}^t(m) = (1-\alpha_t)\frac{1}{m}\sum_{i=1}^m X_i + \alpha_t \hat{P}^{o,t}$    // `regularized least squares minimizer.`

        $\gamma^\star \leftarrow \gamma$-`Selection-Procedure`$(m, \alpha_t, \sigma_t, T, S, A)$

        $\pi^\star_{\hat{P}^t,\gamma^\star} \leftarrow$ `Planning`$(\hat{P}^t(m))$

        **Output:** $\pi^\star_{\hat{P}^t,\gamma^\star}$

    Update $\hat{P}^{o,t+1}$, $\hat{\sigma}_{t+1} \leftarrow$ `Welford's online algorithm`$\left((\hat{\sigma}_o)_t, \hat{P}^{o,t+1}, \hat{P}^{o,t}\right)$    // `meta-update AoM`

    (Eq. 5) and task-similarity parameter.

    Update $\alpha_{t+1} = \frac{1}{\hat{\sigma}_{t+1}{}^2(1+1/t)m+1}$    // `meta-update mixing rate, plug` $\max(\sigma_{S\times A})$

---

**Tasks vary only in certain states and actions.** Thus far, we considered a *uniform* notion of task similarity as Eq. 3 holds for any $(s, a)$. However, in many practical settings the transition distribution might remains the same for most part of the state space but only vary on some states across different tasks. These scenarios are hard to analyse in general because local changes in the model parameters do not always imply changes in the optimal value function nor necessarily modify the optimal policy. Our Theorem 2 still remains valid, but it may not be tight when the meta-distribution has non-uniform noise levels. More precisely Theorem 1 in Appendix D remains locally valid for each $(s, a)$ pair and one could easily replace the uniform $\sigma$ with local $\sigma_{(s,a)}$, but this cannot directly imply a stronger bound on the average planning loss. Indeed, in our experiments, in both `POMRL` and `ada-POMRL` , the parameter $\sigma$ and $\hat{\sigma}$ respectively, are $S \times A$ matrices of state-action dependent variances resulting in state-action dependent mixing rate $\alpha_t$.

# 5    Experiments

We now study the empirical behavior of planning with online meta-learning in order to answer the following questions: **Q1.**Does meta-learning a good initialization of the dynamics model facilitate improved planning accuracy for the choice of $\gamma = \gamma_{\mathtt{eval}}$? (Sec. 5.1) **Q2.**Does meta-learning a good initialization of the dynamics model enables longer planning horizons? (Sec. 5.2) **Q3.**How does performance depend on the amount of shared structure across tasks *i.e.*, $\sigma$? (Sec. 5.3) Source code is provided in the supplementary material.

**Setting:** For each experiment, we fix a mean model $P^o \in \Delta_S^{S \times A}$ (see below how), and for each new task $t \in [T]$, we sample $P^t$ from a Dirichlet distribution[3] centered at $P^o$. As prescribed by theory (see Sec.3.2), we set[4] $\sigma \approx 0.01 \lesssim 1/S\sqrt{m}$ unless otherwise specified (see Q3). Note that $\sigma$ and $\hat{\sigma}$ respectively, are $S \times A$ matrices of state-action dependent variances that capture the directional variance as we used Dirichlet distributions as priors and these have non-uniform variance levels in the simplex, depending on how close to the simplex boundary the mean is located. Aligned with our theory, we use the max of the $\sigma$ matrices resulting in the aforementioned single scalar value. As in Jiang et al. (2015), $P^o$ (and each $P^t$) characterizes a random chain MDP with $S = 10$ states[5] and $A = 2$ actions, which is drawn such that, for each state–action pair, the transition function $P(s, a, s')$ is constructed by choosing randomly $k = 5$ states whose probability is set to 0. Then we draw the value of the $S - k$ remaining states uniformly in $[0, 1]$ and normalize the resulting vector.

## 5.1    Meta-reinforcement learning leads to improved planning accuracy for [$\gamma_{\mathtt{eval}}$]. [Q1.]

We consider the aforementioned problem setting with a total of $T = 15$ closely related tasks and focus on the planning loss gains due to improved model accuracy. We fix $\gamma = \gamma_{\mathtt{eval}}$, a rather naive $\gamma$-Selection-Procedure and show the planning loss of POMRL (Alg. 1) with the following **baselines**: 1) **Oracle Prior Knowledge** knows a priori the underlying task structure ($P^o$, $\sigma$) and uses an estimator (Eq. 4) with exact regularizer $P^o$ and optimal mixing rate $\alpha_t = \frac{1}{\sigma^2(1+1/t)m+1}$, 2) **Without Meta-Learning** simply uses $\hat{P}^t = \bar{P}^t$, the count-based estimated model using the $m$ samples seen in each task, 3) POMRL  (Alg. 1) meta-learns the regularizer but knows apriori the underlying task structure, and 4) ada-POMRL  (Alg. 2) meta-learns not only the regularizer, but also the underlying task-similarity online. The oracle is a strong baseline that provides a minimally inaccurate model and should play the role of an "empirical lower bound". For all baselines, the number of samples per task $m = 5$. Results are averaged over 100 independent runs. Besides, we also propose and empirically validate competitive heuristics for $\gamma$-Selection-Procedure in Sec. 6. Besides, we also run another baseline called Aggregating($\alpha = 1$), that simply ignores the meta-RL structure and just plans assuming there is a single task (See Appendix F.2).

**Inspecting Fig. 4(a)**, we can see that our approach ada-POMRL (green) results in decreasing per-task planning loss as more tasks are seen, and decreasing variance as the estimated model gets more stable and approaches the optimal value returned by the oracle prior knowledge baseline (blue). On the contrary, without meta-learning (red), the agent struggles to cope as it faces new tasks every round, and its performance does not improve. ada-POMRL gradually improves as more tasks are seen whilst adaptation to learned task-similarity on-the-fly which is the primary cause of the performance gap in ada-POMRL and POMRL . Importantly, no prior knowledge about the underlying task relatedness enables a more practical algorithm with the same theoretical guarantees (See Sec. 4). Recall that oracle prior knowledge is a strong baseline as it corresponds to both known task relatedness and regularizer.

## 5.2    Meta-learning the underlying task relatedness enables longer planning horizons. [Q2.]

We run ada-POMRL for $T = 15$ (with $\sigma \approx 0.01$) as above and report planning losses for a range of values of guidance $\gamma$ factors. Results are averaged over 100 independent runs and displayed on Fig. 4(b). We observe

---

[3]The variance of this distribution is controlled by its coefficient parameters $\alpha_{1:S}$: the larger they are, the smaller is the variance. More details on our choices are given in Appendix F.1. Dirichlet distributions with small variance satisfy the high-probability version of our structural assumption 3 for $\sigma = \max_i \sigma_i$

[4]Our priors are multivariate Dirichlet distribution in dimension $S$ so we divide the theoretical rate by $S$ to ensure the max bounded by $1/\sqrt{m}$. See App. F for implementation details.

[5]We provide additional experiments with varying size of the state space in Appendix Fig. F5.

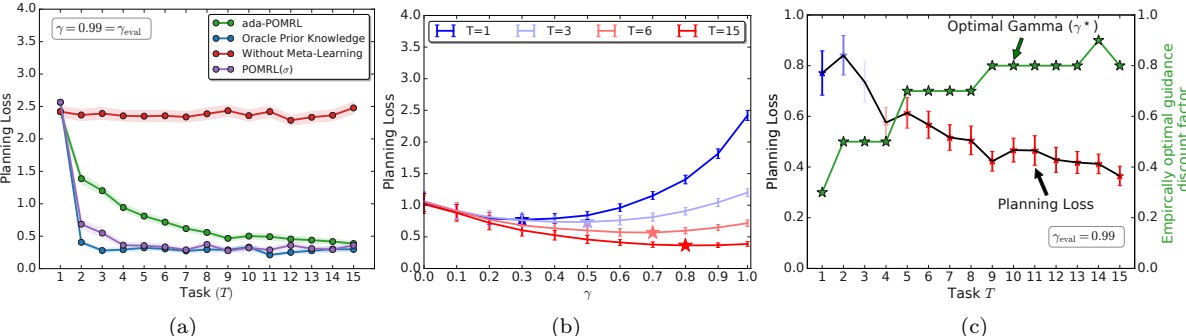

Figure 4: **Planning with Online Meta-Learning. (a) Per-task planning loss** of our algorithms `POMRL` and `ada-POMRL` compared to an Oracle, and Without Meta-learning baselines. All methods use a fixed $\gamma = \gamma_{\text{eval}} = 0.99$. **(b) `ada-POMRL` 's planning loss** decreases as more tasks are seen. Markers denote the $\gamma$ that minimizes the planning loss in respective tasks. Error bars show standard error. **(c) `ada-POMRL` 's empirically optimal guidance discount factor** (right y axis) depicts the effective planning horizon, *i.e.*, one that minimizes the planning loss. Optimal $\gamma$ aka the effective planning horizon is larger with online meta-learning. Planning loss (left y axis) shows the minimum planning loss achieved by the agent in that round $T$. Results are averaged over 100 independent runs and error bars represent 1-standard deviation.

in Fig. 4(b) when the agent has seen fewer tasks $T$, an intermediate value of the discount is optimal, *i.e.*, one that minimizes the task-averaged planning loss ($\gamma^\star < 0.5$). In the presence of strong underlying structure across tasks, **as the agent sees more tasks, the effective planning horizon ($\gamma^\star > 0.7$) shifts to a larger value** - one that is closer to the gamma used for evaluation ($\gamma_{\text{eval}} = 0.99$).

As we incorporate the knowledge of the underlying task distribution, *i.e.*, meta-learned initialization of the dynamics model, we note that the adaptive mixing rate $\alpha_t$ puts increasing amounts of weight on the shared task-knowledge. Note that this conforms to the effect of increasing weight on the model initialization that we observed in Fig. 2. As predicted by theory, the per-task planning loss decreases as $T$ grows and is minimized for progressively larger values of $\gamma$, meaning for longer planning horizons (See Fig. 4(c)). In addition, Appendix Fig. F4 depicts the effective planning horizon individually for `ada-POMRL` , Oracle and without meta learning baselines.

### 5.3 `POMRL` and `ada-POMRL` perform consistently well for varying task-similarity. [Q3.]

We have thus far studied scenarios where the learner can exploit strong task relatedness, *i.e.*, $\sigma \approx 0.01 < 1/(S\sqrt{m})$ (for low data per task *i.e.*, $m = 5$) is small and we now illustrate the other regimes discussed in Section 3.2. We show that our algorithms remain consistently good for all amounts of task-similarity.

We let $\sigma$ vary to cover the **three regimes**: $\sigma \approx 0.01$ corresponding to fast convergence, $\sigma = 0.025$ is in the intermediate regime (needs longer $T$), and $\sigma = 0.047$ is the loosely structured case where we don't expect much meta-learning to help improve model accuracy. The small inset figures in Fig. 5 represent the task distribution in the simplex. In all cases, `ada-POMRL` estimates $\sigma$ online and we report the planning losses for a range of $\gamma$'s. Inspecting Fig. 5, we observe that while in the presence of closely related tasks (Fig. 5(a)) all methods perform well (except without meta-learning). As the underlying task relatedness decreases (for intermediate regime in Fig. 5(b)), both `POMRL` and `ada-POMRL` remain consistent in their performance as compared to the Oracle Prior Knowledge baseline. When the underlying tasks are loosely related (as in Fig. 5(c)), `ada-POMRL` and `POMRL` can still perform well in comparison to other baselines.

Next, we report and discuss the planning loss plot for `ada-POMRL` for the three cases are shown in Figures 5(d), 5(e), and 5(f) respectively. An intermediate value of task-similarity (Fig. 5(e)) still leads to gains, albeit at a lower speed of convergence. In contrast, a large value of $\sigma = 0.047$ indicates little relatedness across tasks resulting in minimal gains from meta-learning here as seen in Fig. 5(f). The learner struggles to learn a good initialization of the model dynamics as there is no natural one. All planning loss curves remain U-shaped and overall higher with an intermediate optimal guidance $\gamma$ value ( 0.5). However, `ada-POMRL` does not do worse overall than the initial run $T = 1$, meaning that while there is not a significant improvement, our method

**Strong Structure**    **Medium Structure**    **Loosely Structured**

(a)                              (b)                              (c)

(d)                              (e)                              (f)

Figure 5: `POMRL` and `ada-POMRL` **are robust to varying task-similarity** $\sigma$ for a small fixed amount of data $m = 5$ available at each round $t \in T$. A small value of $\sigma$ reflects the fact that tasks are closely related to each other and share a good amount of structure whereas a much larger value indicates loosely related tasks (simplex plots illustrate the meta-distribution in dimension 2). In the former case, meta-learning the shared structure alongside a good model initialization leads to most gains. In the latter, the learner struggles to cope with new unseen tasks which differ significantly. Error bars represent 1-standard deviation of uncertainty across 100 independent runs.

does not hurt performance in loosely related tasks[6]. Recall that `ada-POMRL` has no apriori knowledge of the number of tasks ($T$), or the underlying task relatedness ($\sigma$) *i.e.*, adaptation is on-the-fly.

## 6    Adaptation of Planning Horizon $\gamma$

We now propose and empirically validate two heuristics to design an adaptive schedule for $\gamma$ based on existing work (Sec. 6.1) and on our average regret upper bound (Sec. 6.2).

### 6.1    Schedule adapted from Dong et al. (2021) [$\gamma = f(m, \alpha_t, \sigma_t, T)$]

Dong et al. (2021) study a continuous, never-ending RL setting. They divide the time into growing phases $(T_t)_{t \geq 0}$, and tune a discount factor $\gamma_t = 1 - 1/T_t^{1/5}$. We adapt their schedule to our problem, where the time is already naturally divided into tasks: for each $t \geq 0$, we define the phase size $T_t$ and the corresponding $\gamma_t$ as

$$T_0 = m, \quad T_t = \frac{SA}{L}\Big(\underbrace{(1 - \alpha_t)m + \alpha_t m(t - 1)}_{\text{efficient sample size}}\Big), \quad \gamma_t = 1 - \frac{1}{T_t^{1/5}},$$

where $L$ is the maximum trajectory length. The size of each $T_t$, $t \geq 1$, is controlled by an "efficient sample size" which includes a combination of the current task's samples and of the samples observed so far, as used to construct our estimator in `POMRL` .

### 6.2    Using the upper bound to guide the schedule [$\gamma = \min\{1, \gamma_0 + \tilde{\gamma}\}$]

Having a second look at Theorem 2, we see that the r.h.s. is a function of $\gamma$ of the form

$$U : \gamma \mapsto \frac{1}{1 - \gamma_{\text{eval}}} + \frac{1}{\gamma - 1} + C_{m,T,S,A,\sigma,\delta} \frac{\gamma}{(1 - \gamma)^2},$$

---

[6]The theoretical bound may lead to think that the average planning loss is higher due to the introduced bias, but in practice we do not observe that, which means our bound is pessimistic on the second order terms.

where the first term is positive and monotonically decreasing on $(0, \gamma_{\texttt{eval}})$ and the second term is positive and monotonically increasing on $(0, 1)$. We simplify and scale this constant, keeping only problem-related terms: $C_t = (\frac{1}{\sqrt{t}}(\sigma + \frac{1}{\sqrt{m}})/(\sigma^2 m + 1) + \sigma^2 m \frac{1}{\sqrt{m}}/(\sigma^2 m + 1)$, which is of the order of the constant in equation 6. Optimizing $\gamma$ by using the function $U$ with constant $C$ does not lead to a principled analytical value strictly speaking because $U$ is derived from an upper bound that may be loose and may not reflect the true shape of the loss w.r.t. $\gamma$, but we may use the resulting growth schedule to guide our choices online. In general, the existence of a strict minimum for $U$ in $(0, 1)$ is not always guaranteed: depending on the values of $C \approx C_{m,T,S,A,\sigma}$, the function may be monotonic and the minimum may be on the edges. We give explicit ranges in the proposition below, proved in Appendix E.

**Proposition 1.** *The existence of a strict minimum in* $(0, 1)$ *is determined by* $C = C_{m,T,S,A,\sigma,\delta}$ *(which can be computed) as follows:*

$$\tilde{\gamma} = \begin{cases} 0 & \text{if } C \geq 1 \\ 1 & \text{if } C < 1/2 \\ \frac{1-C}{1+C} & \text{otherwise, i.e if } 1/2 < C < 1 \end{cases}$$

We use these values as a guide. Typically, when $T = 1$ and $m$ is small, the multiplicative term $C$ is large and the bound is not really informative (concentration has not happened yet), and $\gamma$ should be small, potentially close to but not equal to zero. As a heuristic, we propose to simply offset $\tilde{\gamma}$ by an additional $\gamma_0$ such that the guidance discount factor is $\gamma = \min\{1, \gamma_0 + \tilde{\gamma}\}$, where $\gamma_0$ should be reasonably chosen by the practitioner to allow for some short-horizon planning at the beginning of the interaction. Empirically, $\gamma_0 = \in (0.25, 0.50)$ seems reasonable for our random MDP setting as it corresponds to the empirical minima on Fig 4(b).

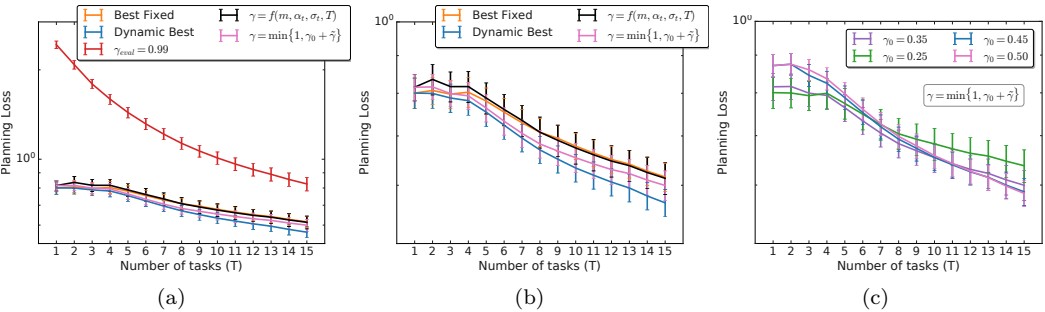

(a)          (b)          (c)

Figure 6: **Adapting the planning horizon during online meta-learning reduces planning loss. (a)** Planning with online-meta learning shows that *all* baselines outperform using a constant discount factor. **(b)** Zoomed in plot of average planning loss over the progression of tasks T shows competitive performance with the proposed schedule of $\gamma = f(m, \alpha_t, \sigma_t, T)$ beating best-fixed as more tasks are seen. The $\gamma$ schedule $\gamma = \min\{1, \gamma_0 + \tilde{\gamma}\}$ using the upper bound as a guidance beats the best-fixed and is very competitive to the dynamic-best baseline. **(c)** Using the upper bound to guide the schedule significantly outperforms $\gamma_{\texttt{eval}}$ and is shown for $\gamma_0 \in (0.25, 0.50)$. Error bars depict 1-standard error for 600 independent runs.

### 6.3 Empirical Validation

Next, we empirically test the proposed schedules for adaptation of discount factors. We consider the setup described in Sec. 5 with 15 tasks in a 10-state, 2-action random MDP distribution of tasks with $\sigma \approx 0.01$. In Fig. 6, we plot the planning loss obtained by `POMRL` with our schedules, a fixed $\gamma_{\texttt{eval}}$ and two strong baselines: *best fixed* which considers the best fixed value of discount over all tasks estimated in hindsight and *dynamic best* which considers the best choice if we had used the optimal $\gamma^\star$ in each round as in Fig. 4(c). It is important to note that *dynamic best* is a lower bound that we cannot outperform.

We observe in Fig. 6(a) that $\gamma_{\texttt{eval}}$ results in a very high loss, potentially corresponding to trying to plan too far ahead despite model uncertainty. Upon inspecting Fig. 6(b), we observe that the proposed $\gamma = f(m, \alpha_t, \sigma_t, T)$ obtains similar performance to *best fixed* and is within the significance range of the lower bound. Our second heuristic, $\gamma = \min\{1, \gamma_0 + \tilde{\gamma}\}$ obtains similarly good performance, as seen in Fig. 6(b). Fig. 6(c) shows the effect of different values of $\gamma_0$ in the prescribed range. These results provide evidence that it is possible to adapt the planning horizon as a function of the problem's structure (meta-learned task-similarity) and sample sizes. Adapting the planning horizon online is an open problem and beyond the scope of our work.

## 7 Discussion and Future Work

We presented connections between planning with inaccurate models and online meta-learning via a high-probability task-averaged regret upper-bound on the planning loss that primarily depends on task-similarity $\sigma$ as opposed to the entire search space $\Sigma$. Algorithmically, we demonstrate that the agent can use its experience in each task *and* across tasks to estimate both the transition model and the distribution over tasks. Meta-learning the underlying task similarity and a good initialization of transition model across tasks enables longer planning horizons.

**Beyond the tabular case:** Function approximation is at the heart of practical RL so a natural question is how to extend our work to parametrized models. For linear MDPs, Müller & Pacchiano (2022) recently derived regret bounds in the fixed-horizon setting for an algorithm using meta-regularizers similar to ours. One question is whether this idea could be extended to infinite horizons and further to non-linear, richer representations. Another, and perhaps deeper question, is around designing and evaluating better planning strategies. Should we revisit such line of work under the light of the planning loss rather than the regret? **On- or Off- Policy Meta-Learning without a simulator:** Realistic problem settings in RL involve using sequentially learnt policies to collect data instead of the simulator. One direction could be to extend our approach to model-based RL algorithms via meta-gradient updates as in ARUBA or MAML, and seek regret guarantees induced by our concentration results. **Non-stationary meta-distribution**: Many real-world scenarios have (slow or sudden) drifts in the underlying distribution itself, e.g. weather. A promising future direction is to consider non-stationary environments where the optimal initialization varies over time.

### Acknowledgments

The authors would like to thank Zheng Wen, Andras Gyorgy, and anonymous reviewers for valuable feedback on a draft of this paper, Sebastian Flennerhag, David Abel, and Benjamin Van Roy for insightful discussions during the course of this project. C.Vernade is funded by the Deutsche Forschungsgemeinschaft (DFG) under both the project 468806714 of the Emmy Noether Programme and under Germany's Excellence Strategy – EXC number 2064/1 – Project number 390727645. CV also thanks the international Max Planck Research School for Intelligent Systems (IMPRS-IS).

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

## A    Additional Related Work

**Discount Factor Adaptation.** For almost all real-world applications, RL agents operate in a much larger environment than the agent capacity in the context of both the computational and memory complexity (e.g. the internet). Inevitably, it becomes crucial to adapt the planning horizon over time as opposed to using a relatively longer planning horizon from the start (which can be both expensive and sub-optimal). This has been extensively studied in the context of planning with inaccurate models in reinforcement learning (Jiang et al., 2015; Arumugam et al., 2018).

Dong et al. (2021) introduced a schedule for $\gamma$ that we take inspiration from in Section 6.1. They consider a 'never-ending RL' problem in the infinite-horizon, average-regret setting in which the true horizon is 1, but show that adopting a different smaller discount value proportional to the time in the agent's life results in significant gains. Their focus and contributions are different from ours as they are interested in asymptotic rates, but we believe the connection between our findings is an interesting avenue for future research.

**Meta-Learning and Meta-RL,** or *learning-to-learn* has shown tremendous success in online discovery of different aspects of an RL algorithm, ranging from hyper-parameters (Xu et al., 2018) to complete objective functions (Oh et al., 2020). In recent years, many deep RL agents (Fedus et al., 2019; Zahavy et al., 2020) have gradually used higher discounts moving away from the traditional approach of using a fixed discount factor. However, to the best of our knowledge, existing works do not provide a formal understanding of why this is helping the agents in better performance, especially across varied tasks. Our analysis is motivated by the aforementioned empirical success of adapting the discount factor. While there has been significant progress in meta-learning-inspired meta-gradients techniques in RL (Xu et al., 2018; Zahavy et al., 2020; Flennerhag et al., 2021), they are largely focused on *empirical analysis* with lot or room for in-depth insights about the source of underlying gains.

## B    Closed-form solution of the regularized least squares

We note that each $\hat{P}$ should be understood as $\hat{P}_{(s,a)}(s')$.

$$\nabla\ell(P|h) = -\frac{2}{m}\sum_{i=1}^{m}(X^i - P) + 2\lambda(P - h)$$

$$\nabla\ell(P|h) = 0 \iff P(1 + \lambda) = \frac{\sum_i X^i}{m} + \lambda h$$

$$\hat{P}_{(s,a)}(s'|h) = \frac{1}{1 + \lambda}\frac{\sum_i X^i}{m} + \frac{\lambda}{1 + \lambda}h \tag{9}$$

$$= \alpha h + (1 - \alpha)\frac{\sum^i X^i}{m} \quad \text{where } \alpha = \frac{\lambda}{1 + \lambda} \tag{10}$$

**Derivation of Mixing Rate $\alpha_t$:**   To choose $\alpha_t$, we want to minimize the MSE of the final estimator.

$$\mathbb{E}_{X\sim P^t}\left(\hat{P}^t - P^t\right)^2 = \mathbb{E}_{X\sim P^t}\left(\alpha_t h_t + (1 - \alpha_t)\bar{P}^t - P^t\right)^2$$

$$= \mathbb{E}_{X\sim P^t}\left(\alpha_t(h_t - P^t) + (1 - \alpha_t)(\bar{P}^t - P^t)\right)^2$$

$$= \alpha_t^2(h_t - P^t)^2 + (1 - \alpha_t)^2\mathbb{E}_{X\sim P^t}\left((\bar{P}^t - P^t)\right)^2$$

where the cross term $2\alpha)t(h_t - P^t)(1 - \alpha_t)\mathbb{E}_{X\sim P^t}E\left[\bar{P}^t - P^t\right] = 0$ since $\mathbb{E}[\bar{P}^t] = P^t$. This is the classic bias-variance decomposition of an estimator and we see that the choice of $h_t$ plays a role as well as the variance of $\bar{P}^t$, which is upper bounded by $1/m$ (because each $X^{i,t}$ is bounded in $(0, 1)$). For instance, for the choice $h_t = P^o$, by our structural assumption 3 we get:

$$\mathbb{E}_{X\sim P^t}\left(\hat{P}^t - P^t\right)^2 \leq \alpha^2\sigma^2 + (1 - \alpha)^2\frac{1}{m},$$

and we minimize this upper bound in $\alpha$ to obtain the mixing coefficient with smallest MSE: $\alpha^* = \frac{1}{\sigma^2 m + 1}$, or equivalently $\lambda^* = \frac{1}{\sigma^2 m}$. Recall this is the within-task estimator's variance where we consider the true $P^o$.

In practice, however, we meta-learn the prior, so for $t > 1$, $h_t = \hat{P}^{o,t} = \frac{1}{t-1} \sum_{j=1}^{t-1} \bar{P}^j$. Intuitively, as $m$ and $t$ grow large, $\hat{P}^{o,t} \to P^o$ and we retrieve the result above (we show this formally to prove Eq. 20 in the proof of our main theorem). To obtain a simple expression for $\alpha_t$, we minimize the "meta-MSE" of our estimator:

$$\mathbb{E}_{P^t \sim P^o} \left( \hat{P}^t - P^t \right)^2 = \alpha_t^2 \mathbb{E}_{P^t \sim P^o} \mathbb{E}_{X \sim P^t} \left( h_t - P^t \right)^2 + (1-\alpha_t)^2 \mathbb{E}_{P^t \sim P^o} \mathbb{E}_{X \sim P^t} \left( (\bar{P}^t - P^t) \right)^2$$

$$\leq \alpha_t^2 \mathbb{E}_{P^t \sim P^o} \left( \frac{1}{t-1} \sum_{j=1}^{t-1} P^j - P^o + P^o - P^t \right)^2 + (1-\alpha_t)^2 \frac{1}{m}$$

$$\leq \alpha_t^2 \sigma^2 (1 + 1/t) + (1-\alpha_t)^2 \frac{1}{m},$$

where in the last inequality, we upper bounded the variance of $\frac{1}{t-1} \sum_{j=1}^{t-1} P^j$ (the "denoised" $\hat{P}^{0,t}$) by $\sigma^2/t$ since each $P^t$ is bounded in $[P^o - \sigma, P^o + \sigma]$ by our structural assumption. Minimizing that last upper bound in $\alpha_t$ leads to $\alpha_t = \frac{1}{(\sigma^2)(1+1/t)m+1} \underset{\to_t}{\leq} \alpha^*$, when $t \to \infty$. This means that the uncertainty on the prior implies that its weight in the estimator is smaller, but eventually converges at a fast rate to the optimal value (when the exact optimal prior is known). This inequality holds with probability $1 - \delta$ because we use the concentration of $\hat{P}^{o,t}$ (see proof of Theorem 17 below)

## C  Online Estimation

**Online Estimation of Prior.**  At each task, the learner gets $m$ interactions per state-action pair. At task $t = 1$, learner can compute the prior based on the samples seen so far, i.e.:

$$\hat{P}_o^{t=1}(s'|s,a) = \frac{\{\sum_{i=1}^m X_i\}_{t=1}}{m}$$

At subsequent tasks,

$$\hat{P}_o^{t=2}(s'|s,a) = \frac{\{\sum_{i=1}^{2m} X_i\}_{t=1:2}}{2m} = \frac{1}{2} \left( \frac{\{\sum_{i=1}^m X_i\}}{m} + \frac{\{\sum_{i=m+1}^{2m} X_i\}}{m} \right)$$

$$= \frac{1}{2} \left( \hat{P}_o^{t=1}(s'|s,a) + \frac{\{\sum_{i=m+1}^{2m} X_i\}}{m} \right)$$

Similarly,

$$\hat{P}_o^{t=3}(s'|s,a) = \frac{\{\sum_{i=1}^{3m} X_i\}_{t=1:3}}{3m} = \frac{\{\sum_{i=1}^{2m} X_i\}_{t=1:2} + \{\sum_{i=2m+1}^{3m} X_i\}_{t=3}}{3m}$$

$$= \frac{1}{3} \left( \frac{2\{\sum_{i=1}^{2m} X_i\}_{t=1:2}}{2m} + \frac{\{\sum_{i=2m+1}^{3m} X_i\}_{t=3}}{m} \right)$$

$$\implies \hat{P}_o^t(s'|s,a) = \frac{1}{t} \left( (t-1)\hat{P}_o^{t-1}(s'|s,a) + \frac{\sum_{i=(t-1)m+1}^{tm} X_i}{m} \right)$$

Therefore,

$$\hat{P}_o^t(s'|s,a) = \left( 1 - \frac{1}{t} \right) \hat{P}_o^{t-1}(s'|s,a) + \left( \frac{1}{t} \right) \frac{\sum_{i=(t-1)m+1}^{tm} X_i}{m} \tag{11}$$

**Online Estimation of Variance.**  Similarly, we can derive the online estimate of the variance:

$$(\hat{\sigma}_o^2)^t = (\hat{\sigma}_o^2)^{t-1} + \frac{(X_{mt} - \hat{P}_o^{t-1})(X_{mt} - \hat{P}_o^t) - (\hat{\sigma}_o^2)^{t-1}}{t} \tag{12}$$

Since the above method is numerically unstable, we will employ Welford's online algorithm for variance estimate.

## D Concentration bounds and Proof of Theorem 2

### D.1 Proof of Theorem 2

We begin the proof by decomposing each term of the loss:

**Lemma 1.** *For a task t denoted by M, and its estimate denoted by $\hat{M}, \forall s \in S$,*

$$V_{P^t,\gamma_{eval}}^{\pi^*_{P^t,\gamma_{eval}}}(s) - V_{P^t,\gamma_{eval}}^{\pi^*_{\hat{P}^t,\gamma}}(s) = \underbrace{\left(V_{P^t,\gamma_{eval}}^{\pi^*_{P^t,\gamma_{eval}}}(s) - V_{P^t,\gamma}^{\pi^*_{P^t,\gamma_{eval}}}(s)\right)}_{A_t} + \underbrace{\left(V_{P^t,\gamma}^{\pi^*_{P^t,\gamma_{eval}}}(s) - V_{P^t,\gamma_{eval}}^{\pi^*_{\hat{P}^t,\gamma}}(s)\right)}_{B_t}$$

We are going to bound each term separately. The term $(A_t)$ corresponds to the bias constant due to using $\gamma$ instead of $\gamma_{eval}$ and was already bounded by Jiang et al. (2015):

**Lemma 2.** *Jiang et al. (2015) For any MDP $\hat{M}$ with rewards in $[0, R_{max}]$, $\forall \pi : S \to A$ and $\gamma \leq \gamma_{eval}$,*

$$V_{P^t,\gamma}^{\pi} \leq V_{P^t,\gamma_{eval}}^{\pi} \leq V_{P^t,\gamma}^{\pi} + \frac{\gamma_{eval} - \gamma}{(1 - \gamma_{eval})(1 - \gamma)} R_{\max} \tag{13}$$

We denote $C(\gamma) = \frac{\gamma_{eval} - \gamma}{(1-\gamma_{eval})(1-\gamma)} R_{\max}$ and notice that $\sum_t A_t/T = C(\gamma)$ so that bounds the first part of the average loss.

To bound the second term $B_t$, we first use Lemma 3 (Equation 18) in Jiang et al. (2015) to upper bound

$$V_{P^t,\gamma}^{\pi^*_{P^t,\gamma_{eval}}}(s) - V_{P^t,\gamma_{eval}}^{\pi^*_{\hat{P}^t,\gamma}}(s) \leq 2 \max_{s \in S, \pi \in \Pi_{R,\gamma}} |V_{P^t,\gamma}^{\pi_{P^t,\gamma_{eval}}}(s) - V_{\hat{P}^t,\gamma_{eval}}^{\pi_{\hat{P}^t,\gamma}}(s)| \tag{14}$$

$$\leq 2 \max_{\substack{s \in \mathcal{S}, a \in \mathcal{A}, \\ \pi \in \Pi_{R,\gamma}}} |Q_{P^t,\gamma}^{\pi_{P^t,\gamma_{eval}}}(s,a) - Q_{\hat{P}^t,\gamma_{eval}}^{\pi_{\hat{P}^t,\gamma}}(s,a)| \tag{15}$$

Using Lemma 4 from Jiang et al. (2015) and noticing that in our setting we do not estimate $R$ so $\hat{R} = R$, $Q_{P^t,\gamma}^{\pi}(s,a) = R(s,a) + \gamma \langle P^t(s,a,;), V_{P^t,\gamma}^{\pi} \rangle$ and $Q_{\hat{P}^t,\gamma}^{\pi}(s,a) = R(s,a) + \gamma \langle \hat{P}^t(s,a,;), V_{\hat{P}^t,\gamma}^{\pi} \rangle$, we have

$$\max_{\substack{s \in \mathcal{S}, a \in \mathcal{A}, \\ \pi \in \Pi_{R,\gamma}}} |Q_{P^t,\gamma}^{\pi_{P^t,\gamma_{eval}}}(s,a) - Q_{\hat{P}^t,\gamma_{eval}}^{\pi_{\hat{P}^t,\gamma}}(s,a)| \leq \frac{1}{(1-\gamma)} \max_{\substack{s \in \mathcal{S}, a \in \mathcal{A}, \\ \pi \in \Pi_{R,\gamma}}} \left| \gamma \langle \hat{P}^t(s,a,;), V_{P^t,\gamma}^{\pi} \rangle - \gamma \langle P^t(s,a,;), V_{P^t,\gamma}^{\pi} \rangle \right| \tag{16}$$

Notice that we are comparing the value functions of two different MDPs which is non-trivial and we leverage the result of Jiang et al. (2015). We refer the reader to the proof of Lemma 4 therein for intermediate steps.

Now summing over tasks, we have

$$\frac{\sum_t (B)_t}{T} \leq \frac{1}{T} \sum_{t=1}^{T} \frac{2}{(1-\gamma)} \max_{\substack{s \in \mathcal{S}, a \in \mathcal{A}, \\ \pi \in \Pi_{R,\gamma}}} \left| \gamma \langle \hat{P}^t(s,a,;), V_{P^t,\gamma}^{\pi} \rangle - \gamma \langle P^t(s,a,;), V_{P^t,\gamma}^{\pi} \rangle \right|$$

$$\leq \frac{2\gamma}{(1-\gamma)} \frac{1}{T} \sum_{t=1}^{T} \max_{\substack{s \in \mathcal{S}, a \in \mathcal{A}, \\ \pi \in \Pi_{R,\gamma}}} \left| \langle \hat{P}^t(s,a,;) - P^t(s,a,;), V_{P^t,\gamma}^{\pi} \rangle \right|$$

$$\leq \frac{2R_{\max}}{(1-\gamma)} \frac{1}{T} \sum_{t=1}^{T} \sum_{s' \in [S]} \max_{\substack{s \in \mathcal{S}, a \in \mathcal{A}, \\ \pi \in \Pi_{R,\gamma}}} \left| \hat{P}^t(s,a,s') - P^t(s,a,s') \right| |V_{P^t,\gamma}^{\pi}|$$

$$\leq \frac{2R_{\max}\gamma}{(1-\gamma)^2} \frac{S}{T} \sum_{t=1}^{T} \max_{s,s' \in \mathcal{S}, a \in \mathcal{A}} \left| \hat{P}^t(s,a,s') - P^t(s,a,s') \right|$$

where we upper-bounded the value function by $R_{\max}/(1-\gamma)$ and one sum over $\mathcal{S}$ by $S \times \max_{s' \in \mathcal{S}} \dots$. Note that this step differs from Jiang et al. (2015) and allows us to boil down to an average (worst-case) estimation error of the transition model. We finally upper bound the r.h.s using Theorem 1 stated and proved below.

**Remark 2.** *In Jiang et al. (2015), the argument is slightly more direct and involves directly controlling the deviations of the scalar random variables $R(s,a) + \gamma \langle \hat{P}^t(s,a,;), V^{\pi}_{\hat{P}^t, \gamma} \rangle$, arguing that it is bounded and centered at $Q^{\pi}_{P^t, \gamma}(s,a)$. This approach is followed by taking a union bound over the policy space $\Pi_{R\gamma}$ and results in a factor $\log(\Pi_{R,\gamma})$ under the square root. We could have followed this approach and obtained a similar result but we made the alternative choice above as we believe it is informative. In our case, this factor is replaced (and upper bounded) by the extra $S$ term. As a result, we lose the direct dependence on the size of the policy class, which is a function of $\gamma$ and should play a role in the bound. In turn, and at the price of this extra looseness, we get a slightly more "exploitable" bound (see our heuristic for a gamma schedule in Section 6). It is easy and straightforward to adapt our concentration bound below to directly bound $R(s,a) + \gamma \langle \hat{P}^t(s,a,;), V^{\pi}_{\hat{P}^t, \gamma} \rangle - Q^{\pi}_{P^t, \gamma}(s,a)$ as in Jiang et al. (2015), and one would obtain a similar bound as Eq. equation 6 without the factor $S$, but with an extra $\log(\Pi_{R,\gamma})$.*

## D.2 Concentration of the model estimator

To avoid clutter in the notation of this section , we drop the $(s,a,s')$ everywhere, as we did in Appendix B above. All definitions of $\hat{P}$ and $\hat{P}_0$ are as stated in the latter section.

**Theorem 1.** *with probability $1-\delta$:*

$$\max_{s,a,s'} |\hat{P}^t - P^t| \leq \frac{1}{\sigma^2 m + 1} \left( \sqrt{\frac{\log(\frac{6T}{\delta}) \log(\frac{TS^2A}{\delta})(\sigma^2 + \frac{\Sigma \log^2(\frac{6T^2}{\delta})}{m})}{T}} + \sigma\sqrt{\frac{\log(\frac{3TS^2A}{\delta})}{T}} + 2\sigma \right)$$

$$+ \frac{\sigma^2 m}{2\sigma^2 m + 1}\sqrt{\frac{\Sigma \log(\frac{3TS^2A}{\delta})}{2m}} \quad (17)$$

For any $t \in [T]$, $s,a,s'$ and $\pi \in \Pi_{R,\gamma}$, define $\hat{P}^{t,*} = \alpha_t P^o + (1-\alpha_t)\bar{P}^t_m$ the optimally regularized estimator (using the true unknown $P^o$ for each $t$). We have

$$\left|\hat{P}^t - P^t\right| \leq |\hat{P}^t - \hat{P}^{t,*}| + |\hat{P}^{t,*} - P_t|$$

$$\leq \underbrace{\alpha_t |\hat{P}^{o,t} - P^o|}_{(A)} + \underbrace{(1-\alpha_t)|\bar{P}^t_m - P^t|}_{(B)} + \underbrace{\alpha_t |P^o - P^t|}_{\leq 2 \cdot \sigma \text{by assum.}} \quad (18)$$

**Bounding Term A**

Substituting the estimator $\hat{P}_t = \alpha_t \frac{1}{m}\sum_i^m X_i + (1-\alpha_t)\hat{P}^t_o$,

$$A \leq \alpha_t \left( |\hat{P}^t_o - \frac{1}{t-1}\sum_{j=1}^{t-1} P^j| + |\frac{1}{t-1}\sum_{j=1}^{t-1} P_j - P_o| \right)$$

$$\leq \frac{1}{\sigma^2 m + 1} \left( \underbrace{|\hat{P}^t_o - \frac{1}{t-1}\sum_{j=1}^{t-1} P^j|}_{(A_1)} + \underbrace{|\frac{1}{t-1}\sum_{j=1}^{t-1} P_j - P_o|}_{A_2} \right)$$

where $\alpha_t$ is simply upper bounded by its initial value $\frac{1}{\sigma^2 m + 1}$ and we introduced the *denoised* (expected) average $\frac{1}{t-1}\sum_{j=1}^{t-1} P^j = \mathbb{E}_{P^1, \dots P^{t-1}}\hat{P}^{o,t}$. Indeed, by assumption, $\mathbb{E}_{P \sim \mathcal{P}} \frac{1}{t-1}\sum_{j=1}^{t-1} P^j = P^o$ and the variance of

this estimator is bounded by $\sigma^2/(t-1)$ by our structure assumption. This allows to naturally bound $A_2$ using Hoeffding's inequality for bounded random variables: with probability at least $1 - \delta/3$,

$$\max_{s,a,s'} A_2 \le \sigma \sqrt{\frac{\log(6S^2 AT/\delta)}{T}} \tag{19}$$

We now bound $A_1$

$$\texttt{A1} = \left| \frac{1}{t-1} \sum_j (\bar{P}_m^j - P^j) \right|$$

We note here that the first term in A1 is indeed a martingale $M_t = \sum_{j=1}^{t-1} Z_j$, where $Z_j = \bar{P}_m^j - P^j$, such that each increment is bounded with high probability: for each $j$, $|Z_j| \le c_j$ w.p $1 - \frac{\delta}{6}$, where $c_j = \sqrt{\frac{\Sigma}{m} \log(\frac{6T^2}{\delta})}$. Moreover, the differences $|Z_j - Z_{j+1}|$ are also bounded with high probability:

$$|Z_j - Z_{j+1}| \le |P^j - P^{j+1}| + |\bar{P}^j - \bar{P}^{j+1}| < 2\sigma + 2c_j = D_j = 2 \left( \sigma + \frac{\sqrt{\Sigma} \log(\frac{6T^2}{\delta})}{\sqrt{m}} \right)$$

Then by (Tao & Vu, 2015, Prop. 34), for any $\epsilon > 0$,

$$P\left( \left| \frac{M_t}{t-1} \right| \ge \frac{\epsilon}{t-1} \sqrt{\sum_{j=1}^{t-1} D_j^2} \right) \le 2\exp(-2\epsilon^2) + \sum_{j=1}^{t-1} \frac{\delta}{6T^2}$$

Choosing $\epsilon = \sqrt{\frac{1}{2} \log(\frac{12T}{\delta})}$, we get

$$P\left( \left| \frac{1}{t-1} \sum_{j=1}^{t-1} \bar{P}_m^j - P_j \right| \ge \sqrt{\frac{(\sigma + \frac{\sqrt{\Sigma} \log(\frac{6T^2}{\delta})}{\sqrt{m}})^2 \log(\frac{6T}{\delta})}{T}} \right) \le \frac{\delta}{6T} + \frac{\delta}{6T} = \frac{\delta}{3T}$$

With a union bound as before, we get that with probability at least $1 - \delta/3$,

$$A_1 \le \sqrt{\frac{\log(\frac{6T}{\delta}) \log(\frac{TS^2 A}{\delta})(\sigma^2 + \frac{\Sigma \log^2(\frac{6T^2}{\delta})}{m})}{T}} \tag{20}$$

because $(\sigma + \sqrt{\frac{\Sigma}{m}})^2 \ge \sigma^2 + \frac{\Sigma}{m}$.

By combining equation 20 and equation 19, we get:

$$\max_{s,a,s'} \alpha_t |P^{o,t} - P^o| \le \frac{1}{\sigma^2 m + 1} \left( \sqrt{\frac{\log(\frac{6T}{\delta}) \log(\frac{TS^2 A}{\delta})(\sigma^2 + \frac{\log^2(\frac{6T^2}{\delta})}{m})}{T}} + \sigma \sqrt{\frac{\log(3S^2 AT/\delta)}{T}} \right) \tag{21}$$

**Bounding Term B**

Term B is simply the concentration of the average of bounded variables $\bar{P}_m^t = \frac{1}{m} \sum_i X_i$, whose variance is bounded by 1. So by Hoeffding's inequality, and a union bound, with probability at least $1 - \delta/4$

$$\max_{s,a,s'} |\bar{P}_m^t - P^t| \le \sqrt{\frac{\Sigma \log(4TS^2 A/\delta)}{2m}}$$

To bound term B, it remains to upper bound $1 - \alpha_t$ for all $t \in [T]$:

$$1 - \alpha_t = \frac{\sigma^2(1 + \frac{1}{t})m}{\sigma^2(1 + \frac{1}{t})m + 1} \leq \frac{\sigma^2 m}{2\sigma^2 m + 1}$$

We get that with probability $1 - \delta/3$

$$\max_{s,a,s'}(B) \leq \frac{\sigma^2 m}{2\sigma^2 m + 1} \sqrt{\frac{\Sigma \log(3TS^2 A/\delta)}{2m}} \tag{22}$$

**Combining all bounds**

To conclude, we combine the bounds on the terms in equation 18, replacing with equation 21, equation 22, and with a union bound, we get that with probability $1 - \delta$,

$$\max_{s,a,s'} |\hat{P}^t - P^t| \leq \frac{1}{\sigma^2 m + 1} \left( \sqrt{\frac{\log(\frac{6T}{\delta}) \log(\frac{TS^2 A}{\delta})(\sigma^2 + \frac{\Sigma \log^2(\frac{6T^2}{\delta})}{m})}{T}} + \sigma \sqrt{\frac{\log(3S^2 AT/\delta)}{T}} + 2\sigma \right)$$
$$+ \frac{\sigma^2 m}{2\sigma^2 m + 1} \sqrt{\frac{\Sigma \log(3TS^2 A/\delta)}{2m}} \tag{23}$$

**Discussion**

The bound has 4 main terms respectively in $\tilde{O}(\sqrt{\frac{1}{mT}})$, $\tilde{O}(\sqrt{\frac{1}{T}})$, $\tilde{O}(\frac{1}{m})$ and $\tilde{O}(\sqrt{\frac{1}{m}})$, all scaled by some factor depending on $\sigma^2$ and $m$. A first remark is that when $m$ is large and $T = 1$, the last part in $\tilde{O}(\sqrt{\frac{1}{m}})$ dominates due to the factor $\frac{\sigma^2 m}{\sigma^2 m + 1} \to 1$, while the coefficient of the first two terms goes to 0 fast (in $1/(\sigma^2 m)$).

## E   Proof of Proposition 1

We study the function $U$ defined by

$$U : \gamma \mapsto \frac{1}{1 - \gamma_{\texttt{eval}}} + \frac{1}{\gamma - 1} + C_{m,T,S,A,\sigma,\delta} \frac{\gamma}{(1 - \gamma)^2} \,,$$

where $\gamma_{\texttt{eval}}$ is a fixed constant and $C := C_{m,T,S,A,\sigma,\delta}$ is seen as a parameter whose value controls the general "shape" of the function. We differentiate with respect to $\gamma$:

$$\frac{dU}{d\gamma} = -\frac{-C(\gamma + 1) + (1 - \gamma)}{(1 - \gamma)^3}.$$

We see that the sign of the derivative is affected by the value of the parameter $C$:

- If $\forall \gamma \in (0,1), -C(\gamma + 1) + (1 - \gamma) > 0$ then $U$ is monotonically decreasing on $(0,1)$ and the minimum is reached for $\gamma = 1$,

$$\forall \gamma \in (0,1), \ -C(\gamma + 1) + (1 - \gamma) > 0 \iff -2C + 1 > 0 \iff C < 1/2.$$

- Similarly, if $C$ is really large, $U$ may be monotonically increasing on $(0,1)$:

$$\forall \gamma \in (0,1), \ -C(\gamma + 1) + (1 - \gamma) < 0 \iff C \geq 1;$$

- Finally, if $C \in (1/2, 2)$, the minimum exists inside $(0,1)$ and is reached for

$$-C\gamma - C + 1 - \gamma = 0 \iff \gamma = \gamma^* = \frac{1 - C}{1 + C}$$

# F Experiments: Implementation Details, Ablations & Additional Results

## F.1 Implementation details

We consider a Dirichlet distribution of tasks such that all tasks $t \in [T]$, $P^t \sim \mathcal{P}$ are centered at some fixed mean $P^o \in \Delta_S^{S \times A}$ as shown in Figure F1. The mean of the task distribution $P^o$ is chosen as a sampled random MDP and variance of this distribution is determined such that $\|P_{s,a}^t - P_{s,a}^o\|_\infty \leq \sigma < 1$. Next, we compute the variance of this distribution $\sigma_i = \frac{\tilde{\alpha}_i(1-\tilde{\alpha}_i)}{\alpha_0+1}$, where $\tilde{\alpha}_i = \frac{\alpha_i}{\alpha_0}$ and $\alpha_0 = \sum_i^S \alpha_i$.

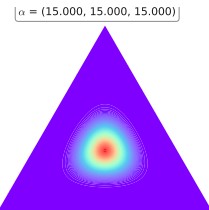

Figure F1: **Dirichlet Task Distribution** for $S = 3$ states, with $Dir(\alpha)$ where $\alpha = [15, 15, 15]$, resulting in our task-similarity measure approximately to be $\sigma = 0.0129$.

## F.2 Ablations

We also run ablations with **Aggregating($\alpha = 1$)**, a naive baseline that simply ignores the meta-RL structure and just plans assuming there is a single task. We observe in Fig. F2 the aggregating baseline works at-par with our method POMRL which is intuitive when the tasks are strongly related to each other in this case. However, as the underlying task structure decreases, we note that Aggregating($\alpha = 1$) as though it is one single task is problematic and suffers from a non-vanishing bias due to which for each new task there is on average an error which does not go to zero. More importantly, the Aggregating($\alpha = 1$) baseline cannot have the same guarantees as POMRL and ada-POMRL .

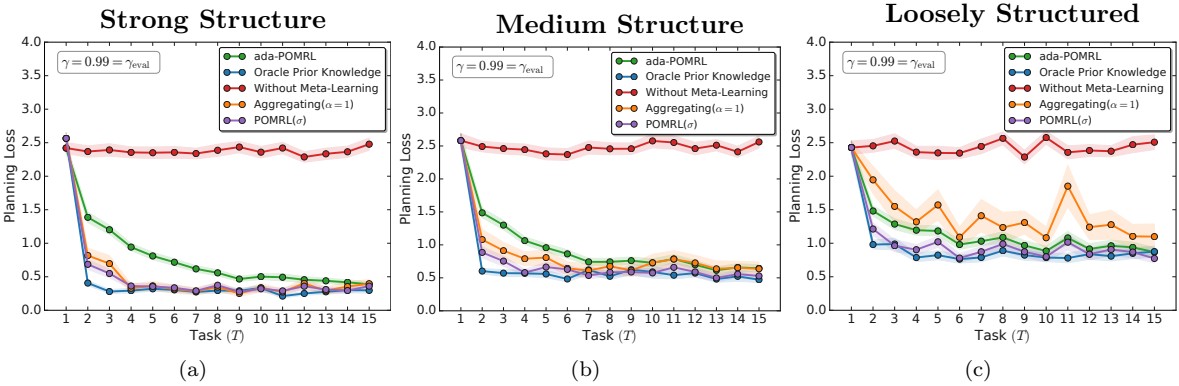

Figure F2: **Ablations for Efficacy of POMRL and ada-POMRL for varying task-similarity.** depicts the effect of the task-similarity parameter $\sigma$ for a small fixed amount of data $m = 5$ available at each round. We run another baseline called Aggregating (orange) that simply ignores the meta-RL structure and acts as if it is all one single task. In the presence of strong structure, meta-learning the shared structure alongside a good model initialization leads to most gains and even naively aggregating the tasks transitions might seem to work well. However, such a naive method is not reliable as the underlying task similarity decreases - the learner struggles to cope with new unseen tasks which differ significantly and the planning loss doesn't improve. Error bars represent 1-standard deviation of uncertainty across 100 independent runs.

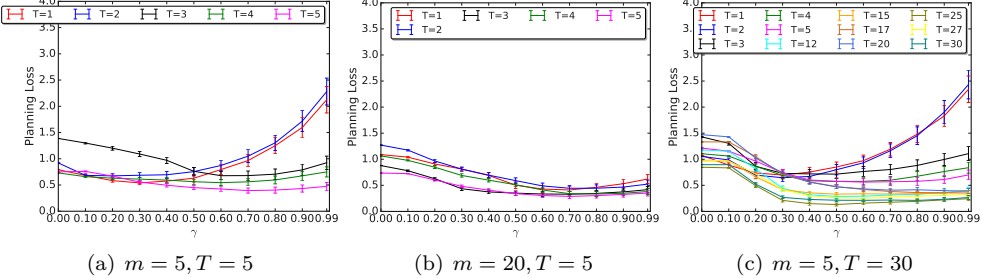

(a) $m = 5, T = 5$       (b) $m = 20, T = 5$       (c) $m = 5, T = 30$

Figure F3: **Effect of m and T on Average Regret Upper Bound on Planning:** for a fixed value of task similarity $\sigma$, depends on the number of samples per task $m$ and the number of tasks $T$. (a) For $\mathbf{m = T}$, smaller loss is obtained with very small discount factor. This implies that with a lot of uncertainty it is not interesting to plan far too ahead, (b) For $\mathbf{m >> T}$, each task has enough samples to inform itself resulting in slightly larger effective discount factors. Not a lot is gained in this scenario from meta-learning, (c) $\mathbf{m \ll T}$ is the most interesting case as samples seen in each individual task are very limited due to small $m$. However, the number of tasks are much more resulting in huge gains from leveraging shared structure across tasks.

### F.3 Additional Experiments

We examine more properties of `ada-POMRL` , namely **Effect of $m$, and $T$ on Planning Loss** in Fig. F3, **Individual Baseline's Performance** in Fig. F4, and **Varying State Space $|S|$, $m$, and $T$** in Fig. F5.

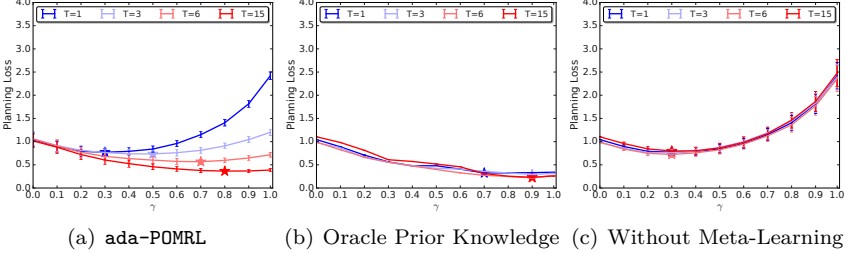

(a) `ada-POMRL`     (b) Oracle Prior Knowledge     (c) Without Meta-Learning

Figure F4: **Planning with Online Meta Learning - Baselines.** (a) `ada-POMRL` . Meta updates include learning $P_o$, $\sigma$, $\alpha$ as a function of tasks. (b) **Oracle Prior Knowledge** considers the optimal $\alpha$, true mean of the task distribution $P_o$ and actual underlying task similarity $\sigma$ as known apriori, (c)**Without Meta-Learning** estimates the transition kernel in each round $T$ without any meta-learning. All baselines are obtained with $T = 15$ tasks and $m = 5$ samples per task.

### F.4 Reproducibility

We follow the reproducibility checklist by Pineau (2019) to ensure this research is reproducible. For all algorithms presented, we include a clear description of the algorithm and source code is included with these supplementary materials. For any theoretical claims, we include: a statement of the result, a clear explanation of any assumptions, and complete proofs of any claims. For all figures that present empirical results, we include: the empirical details of how the experiments were run, a clear definition of the specific measure or statistics used to report results, and a description of results with the standard error in all cases.

### F.5 Computing and Open source libraries.

All experiments were conducted using Google Colab instances[7].

---

[7]https://colab.research.google.com/

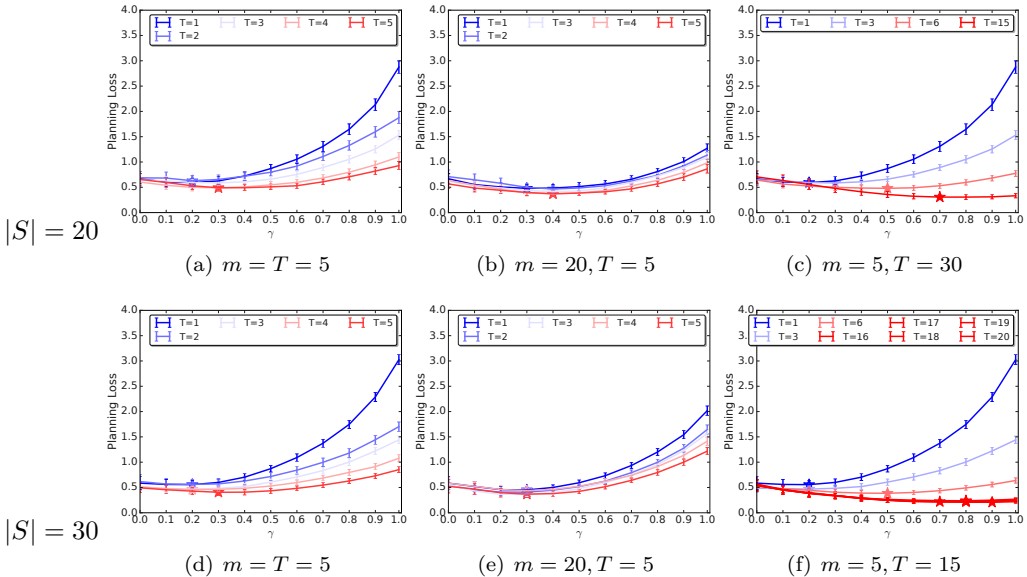

Figure F5: **Varying the size of state-space $S$, number of samples per task $m$, and number of tasks $T$, on Task-averaged Regret Upper Bound on Planning:** for a fixed value of task similarity $\sigma$, We note that despite larger state-space we observe the same effect i.e. (a,d,g) For **m = T**, smaller loss is obtained with very small discount factor i.e. a lot of uncertainty and inability to plan far too ahead, (b,e,h) For **m >> T**, each task has enough samples to inform itself resulting in slightly larger effective discount factors. Not a lot is gained in this scenario from meta-learning. (c,f,i) **m ≪ T** is the most interesting case as samples seen in each individual task are very limited due to small $m$. Meta-learning has most significant gains in this case by leveraging the structure across tasks. Results are averaged over 20 independent runs and error bars represent 1-standard deviation.

