# OpenReview forum: "POMRL: No-Regret Learning-to-Plan with Increasing Horizons"
_TMLR — Accepted by TMLR_

### Review · Reviewer_pog9 · 2023-01-24

**Summary Of Contributions:**

In this work, the authors study online meta-learning. Specifically, they study a problem in which an agent sequentially tackles related tasks. The agent uses data acquired from previous tasks to learn an initial guess of the current task model. The proposed analysis is inspired by the Average Regret-Upper-Bound Analysis (ARUBA) framework, adapted to the meta-learning scenario. Regret upper bounds are derived and numerical results on a synthetic experiment are provided. Source code for the experimental evaluation is provided.

**Audience:**

Yes

**Broader Impact Concerns:**

None.

**Claims And Evidence:**

Yes

**Requested Changes:**

I find the paper to be of reasonable quality. I'd suggest some of the modifications mentioned in "Weaknesses", but overall I found this work to be well-written and worthy of publication.

**Strengths And Weaknesses:**

Strengths.

The paper is well written and clearly positioned with respect to the state of the art. The topic addressed is relevant to the community and the results introduced in the paper appear to be correct.

Weaknesses.

Some of the assumptions in the paper could use some stronger motivation. Not saying that they are unreasonable, but it would help to clarify their need to the reader and how to overcome them in practice (i.e., assumption of known rewards, access to simulator of transitions, etc).

Numerical evaluations are provided for synthetic data. It would be interesting to see the method applied to real datasets instead of a synthetic experiment.

---

> ### Author Response · Authors · 2023-03-30
> **Rebuttal Response by Authors**
>
> Thank you for your comments and feedback. We have made the requested changes that are within the scope of this work. Please see the full list in common response.
>
> **Re motivation.** From common response "We acknowledge our assumptions but also note that this is an intentional choice to establish the first line of results in this under-explored planning in meta RL. Besides, we note that the task-similarity assumption (Sec 3.1.1) is a standard statistic theory assumption where we only require that P has bounded support and not necessarily a Gaussian in the Regularized Least Squares approach [Baxter, 2000]. Similarly, it is typical of RL theory methods to assume that we have access to the state-action-next-state transition tuple via access to a simulator (Kakade, 2013) or through batch data (Munos et al., 2008; Farahmand et al., 2010;
> Chen et al., 2019). In practise, access to a simulator can be relaxed in real-world applications by directly acting in the world, e.g. a robotic arm observes a state (through a camera or a sensor), takes an action, and observes the next state. Modelling reward functions is also feasible e.g. from human preferences or large models"
>
>
> **Re synthetic data.** We agree that it would be interesting to see the method applied to real world applications such as robotics. However, we note that there are multi-faceted challenges within RL theory itself and we chose to address this in a tabular toy domain for corroborating the theoretical claims we establish in this work. We have added in the discussion how our work can be extended to parameterized models. Please see common response to all and the revised draft.

---

### Review · Reviewer_7X2a · 2023-02-07

**Summary Of Contributions:**

The paper develops model-based methods in meta-reinforcement learning. Its contributions are fourfold:
- they conceptualize and formalize the problem of planning under model uncertainty
- they prove regret bounds in the above formalization
- they propose an algorithm to solve the meta-reinforcement learning setting. In particular, they present versions that learn the task-similarity parameter and the planning horizon


**Audience:**

Yes

**Claims And Evidence:**

Yes

**Requested Changes:**

Discuss the expected impact of the results on the field. For example, running empirical evaluations in a non-tabular setting (even simplified) might, in my opinion, significantly strengthen the paper.


**Strengths And Weaknesses:**

Strengths:
- the paper is well-executed. The write-up is very clean, the structure is correct, and the research questions are well-stated
- the problem studied in the paper, planning in a meta-reinforcement learning setting, is of significant
- Conceptualization and formalization of the problem, even if unsurprising, is valuable for the community
- The provided regret bounds further solidify the theoretical contribution
- The experimental part nicely illustrates the theoretical part

Weaknesses:
- The whole analysis is concerned with the tabular setting, and then even is mostly useful, then the 'size' of the environment is modest. This makes it hard to assess what is the long-term significance. I understand that providing any theory for the case when approximators are used is probably very hard. Nevertheless, even a simple empirical evaluation would be of value.

---

> ### Author Response · Authors · 2023-03-30
> **Rebuttal Response by Authors**
>
> Thank you for your comments and feedback. We have made the requested changes that are within the scope of this work. Please see the full list in common response.
>
> **Broader impact of the work.** We first note that of the sizable literature on online learning with linear function approximation theory (Jiang et al., 2017; Du et al., 2019b; Jin et al.,
> 2020; Wang et al., 2019; Yang and Wang, 2019; Ayoub et al., 2020; Modi et al., 2020; Wang et al., 2020b; Zanette et al., 2020, Weisz et al., 2021), none of them concerns with online meta-learning and planning all at the same time. Due to the multiple dimensions of the problem formulation at hand i.e. theory, tabular/function approximation, planning, and meta learning, we chose to work with tabular setting to derive high probability bounds. We believe our work is a crucial first step in establishing the problem of planning with online meta-RL and provides the fundamental ground to develop further. To alleviate concerns about tabular setting, we have added concrete discussion how our work can be extended to function approximation. Please see common response to all and the revised draft.
>
> **Regarding "the 'size' of the environment.** While the main paper only has experiments for |S|=10, we also investigated the impact of varying the size of state-space S, number of samples per task m, and number of tasks T, on the Task-averaged Regret Upper Bound for Planning. Figure F5 in the supplementary material depicts that despite relatively larger state-space (|S|=20, |S|=30) we observe the same effect further corroborating our theoretical claims.

---

### Review · Reviewer_C8FM · 2023-03-19

**Summary Of Contributions:**

This paper considers an environment where a reinforcement learning (RL) agent faces a sequence of related tasks. The objective is to use an online meta-learning approach that learns and exploits the underlying similarities across tasks to effectively adapt to every new task (through planning in a model-based approach). The paper proposes a practical algorithm for the stated objective and theoretically provides a task-averaged regret upper-bound on the planning loss. This bound shows that the regret decreases with the number of tasks and the associated task similarity. Further, the paper provides two heuristics for selecting slowly increasing discount factors, in this online meta-learning approach. Empirically the paper demonstrates that meta-learning an initialization of the transition model and a distribution (underlying similarity) across tasks leads to improved accuracy for planning in each new (unseen) task, and planning ahead (i.e., planning using long horizons).

**Audience:**

Yes

**Broader Impact Concerns:**

No concerns.

**Claims And Evidence:**

Yes

**Requested Changes:**

I only have some minor comments in the paper. I encourage the authors to incorporate as many of these as possible. I would like to recommend acceptance of the paper and the requested changes here are not critical to changing my recommendation.

1) Assumptions: The paper rolls in several assumptions into the text at various places, which hurts the flow of reading and causes some misunderstandings. For example, in Section 3.2, the authors state that "we shall assume throughout that the rewards are known", and in Section 2, "we assume that we have access to a simulator". I recommend that the authors list their set of assumptions formally early on and refer back to these assumptions in the appropriate places in the text.

2)  Throughout the paper it would be useful to have some real-world examples that motivate the problem setting. I think there can be some immediate applications to domains such as robotics. The authors can use such examples to provide intuition in several places in the paper.

3) I had trouble understanding Equation 4 and connecting back to the text around it. I would encourage the authors to explain each term and variable used in this equation precisely along with the necessary intuitions.

4) In several places, the authors use the term "task similarity structure" or "underlying structure". The meaning of "structure" here was unclear to me. It would be helpful to define/introduce this term early on (or during the first usage) in the paper.

5) In Figure 1, I did not get the meaning of the (blue) circles and arrows in the left-most figure. It will be helpful for the authors to clarify that in the description of this figure.

6) Minors: Section 2: consequently to define -> consequently we define; Section 4: comes comes -> comes; Section 4: will gives a -> will give a; Section 5: several places: dynamics model -> the dynamics model; section 5.1: estimator -> an estimator.



**Strengths And Weaknesses:**

Strengths: The paper is well-written and clear. The paper presents its contributions clearly and does a nice job in situating the work in literature. The problem addressed is important and fundamental with a large set of possible real-world applications (though it would be nice for the authors to discuss some of them, preferably right in the introduction where they are motivating the work).

Weaknesses: I think the paper has some very stringent assumptions that reduces the scope of the work. Assumptions of stationary task distribution, similarity of tasks, the tabular setting, and assumptions of known rewards (in several places) are very limiting. The authors could have provided some detailed discussions regarding relaxing some of these assumptions at-least empirically. Nonetheless, I agree that this is a novel theoretical effort and such limiting assumptions may be required in establishing the problem framework and providing the first line of results.

---

> ### Author Response · Authors · 2023-03-30
> **Rebuttal Response by Authors**
>
> Thank you for your valuable feedback and insights. Please see the list of changes in the common response, we have addressed all the requested changes including minor edits.
>
> **Re Stringent Assumptions:** From common response: "We acknowledge that our assumptions are strong but also note that this was an intentional choice to establish the first line of results in this under-explored planning in meta RL. Besides, we note that the task-similarity assumption (Sec 3.1.1) is a standard statistic theory assumption where we only require that P has bounded support and not necessarily a Gaussian in the Regularized Least Squares approach [Baxter, 2000]. Similarly, it is typical of RL theory methods to assume that we have access to the state-action-next-state transition tuple via access to a simulator (Kakade, 2013) or through batch data (Munos et al., 2008; Farahmand et al., 2010;
> Chen et al., 2019). In practise, access to a simulator can be relaxed by directly acting in the world, e.g. a robotic arm observes a state (through a camera or a sensor), takes an action, and observes the next state. Modelling reward functions is also feasible e.g. from human preferences or large models." We have listed the set of assumptions made formally in Section 3.1 and refer back to them as and when needed. We hope this improves the flow of the reading and overall comprehension. See Section 3.1.
>
> **Motivation.** In many real world scenarios such as robotics, it is required to be responsive to changes in the environment and, at the same time, to be robust against perturbation inherent in the environment and their decision making. We have added this motivation around the text in both the introduction and Section 3.1.1 where we formally define the structural assumption across tasks. Please also see common response.
>
> **Equation 4.** We have simplified Equation 4 including explanation of the more complex terms connecting to the text already around it.
>
> **Re ``underlying structure".** We have clarified this term during the first usage in the introduction. Moreover, to further improve comprehension, we have used the term `relatedness' as opposed to structure where feasible. By underlying structure, we refer to how the tasks are related to each other. More specifically, how the transition dynamics across tasks are related.
>
> **Re Figure 1.** we have added to the caption that the blue circles indicate each task $P^{t}$ where the diameter of each circle indicates the variance in the transition dynamics of that task ($v^2$). In left most figure, the red circle diameter represents the variance parameter $\sigma$ also known as the measure of task-similarity centered at mean $P^o$. The arrow is simply pointing to the mean of a Gaussian meta learned model. Please see revised caption of Figure 1.

---

### Decision · Action_Editors · 2023-05-05

**Recommendation:** Accept as is

**Comment:**

The paper proposes a new model-based meta-learning technique for RL.  While the technique is only demonstrated in the context of synthetic tabular domains, it lends itself to a strong theoretical analysis.  More specifically, regret bounds are provided under syutable assumption.  The applicability of those assumptions was questioned by the reviewers, but they are common for a theoretical analysis.  Overall, this work makes several important theoretical contributions that advance the state of the art of meta RL and our understanding of the various factors that influence performance in meta-RL.

**Audience:**

This paper will be of interest to the reinforcement learning community.

**Claims And Evidence:**

The paper proposes a new model-based meta-learning technique for RL.  A strong theoretical analysis is provided with regret bounds.  An empirical proof of concept with synthetic tabular domains provide additional support.